# ADOPD: A LARGE-SCALE DOCUMENT PAGE DECOMPOSITION DATASET

**Jiuxiang Gu**[1]* **Xiangxi Shi**[2] **Jason Kuen**[1] **Lu Qi**[3] **Ruiyi Zhang**[1] **Anqi Liu**[4]
**Ani Nenkova**[1] **Tong Sun**[1]
[1]Adobe Research  [2]Oregon State University  [3]UC, Merced  [4]Johns Hopkins University

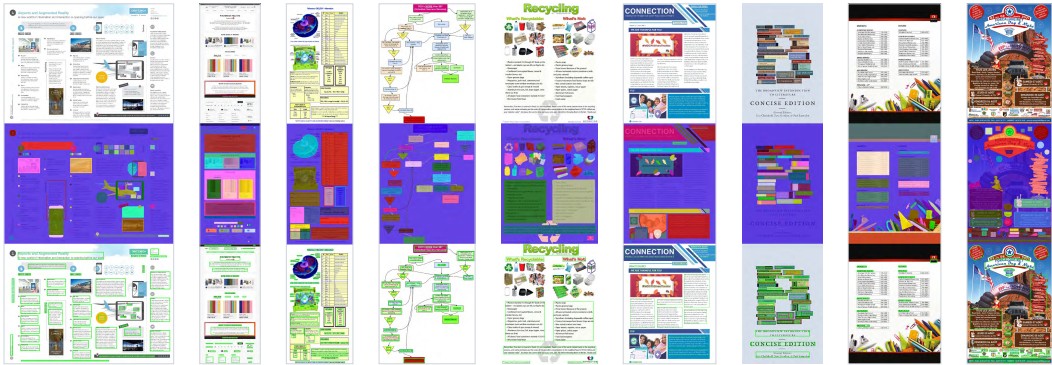

Figure 1: Overview of the ADOPD dataset showcasing densely annotated images of various document types and layouts. Each column presents the original image alongside visual entity masks and annotations of text bounding boxes, organized from top to bottom.

## ABSTRACT

Research in document image understanding is hindered by limited high-quality document data. To address this, we introduce ADOPD, a comprehensive dataset for document page decomposition. ADOPD stands out with its data-driven approach for document taxonomy discovery during data collection, complemented by dense annotations. Our approach integrates large-scale pretrained models with a human-in-the-loop process to guarantee diversity and balance in the resulting data collection. Leveraging our data-driven document taxonomy, we collect and densely annotate document images, addressing four document image understanding tasks: Doc2Mask, Doc2Box, Doc2Tag, and Doc2Seq. Specifically, for each image, the annotations include human-labeled entity masks, text bounding boxes, as well as automatically generated tags and captions that have been manually cleaned. We conduct comprehensive experimental analyses to validate our data and assess the four tasks using various models. We envision ADOPD as a foundational dataset with the potential to drive future research in document understanding[1].

## 1 INTRODUCTION

Document understanding has been invigorated by the introduction of large-scale document datasets (Zhong et al., 2019; Mondal et al., 2020; Cheng et al., 2023), supporting a variety of document-related tasks (Mathew et al., 2021; Mathur et al., 2023). However, document datasets still fall short compared to data resources in more established fields (Gu et al., 2018), in which advances have been so great that models and solutions can be incorporated in real-world applications. A case in point is the field of image decomposition, where progress was fueled by datasets like MSCOCO (Lin et al., 2014) and Pascal VOC (Everingham et al., 2010). Building a document page decomposition dataset of comparable quality is essential to advance document understanding research.

---

*Correspondence to: jigu@adobe.com
[1]Project page: https://adopd2024.github.io

We construct ADOPD by addressing two important questions: *(1) How do we gather document data, and what types of documents should be included in the dataset?* Table 1 compares ADOPD with earlier datasets for document layout analysis (Mondal et al., 2020; Smock et al., 2022; Landeghem et al., 2023; Saad et al., 2016)[2]. Most datasets are sourced from PDFs, with limited document types. Models trained on such homogeneous data are unlikely to perform well on different types of documents, so a top prority when collecting ADOPD is to maximize the diversity of documents types in it. *(2) What elements should be annotated in document images for page decomposition?* Documents, with their varied forms, can be interpreted differently based on an individual's background. Document understanding encompasses intricacies such as visuals, text, and layout. For instance, a poster with a form may visually seem like a form, yet its text could classify it as a science or education book. The complex nature of document data poses challenges in hierarchically structuring it, a critical aspect for successful vision datasets like ImageNet and MSCOCO. Meanwhile, accurately describing the content of documents is highly valuable, but it is also more challenging than natural image captioning.

We explore the fundamental question: *How can we obtain a reasonable taxonomy of document types*? Pre-defining a fixed taxonomy solely based on human knowledge is not practical. Instead we assume an open taxonomy and make use of a data-driven taxonomy discovery method, gradually assembling the taxonomy through large-scale data exploration. Relying solely on manual annotation of document types, which requires reading and understanding the document content, is also not practical. Therefore, we leverage the powerful zero-shot capabilities of large pretrained models such as CLIP (Radford et al., 2021) and

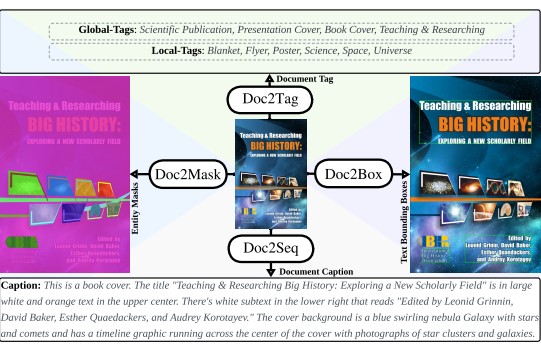

Figure 2: ADOPD page decomposition illustration.

Large Language Models (LLMs) (Floridi & Chiriatti, 2020) to assist in data selection and analysis. We couple the language model with methods for out-of-distribution (OOD) detection (Gu et al., 2023) for outlier data selection, complemented by a human-in-the-loop (HITL) approach to achieve data diversity. Each ingredient in our proposed approach—LLM, OOD and HITL—is imperfect, but together they support the selection and annotation of diverse data at scale, within a reasonable budget.

Fig. 1 illustrates the diverse document types in ADOPD, comprising visually and textually rich documents, posing an annotation challenge despite its advantage. For visually rich documents like posters and diagrams, entity masks capture relationships between visual elements effectively. Conversely, for text-rich documents such as letters and articles, text bounding boxes are more suitable for marking key textual elements. To accomodate both types, we segment

Table 1: Comparison of document datasets.

| Dataset | Year | Size | Anno (Type) | Category |
|---------|------|------|-------------|----------|
| PubLayNet | 2019 | 360K | Bbox (💻) | 📄 (1) |
| DocBank | 2020 | 500K | Bbox (💻) | 📄 (1) |
| IIIT-AR-13K | 2020 | 13k | Bbox (👥) | 🖼 (1) |
| DocLayNet | 2022 | 80.9k | Bbox (👥) | 📄 (6) |
| M⁶Doc | 2023 | 9.1k | Bbox (👥) | 📄🖼 (7) |
| ADOPD (**Ours**) | 2024 | 120k | Polygon (👥)
Text Bbox (👥)
Caption (🤖+👥)
Tag (🤖+👥) | 🖼 (>1000) |

each document into entity masks and text regions, and provide two types of descriptive labels for each document image. Fig. 2 showcases the four document page tasks: entity segmentation (Doc2Mask), text detection (Doc2Box), tagging (Doc2Tag), and captioning (Doc2Seq).

In sum, ADOPD is a large-scale diverse document page decomposition and understanding dataset, designed to support future research in document domain. In this paper, we:

- present ADOPD, comprehensive dataset for document page decomposition, encompassing four distinct tasks: Doc2Mask, Doc2Box, Doc2Seq, and Doc2Tag.
- propose a data-driven approach for constructing document taxonomies during data collection and safeguard the ADOPD through outlier detection and human-in-the-loop.
- conduct extensive experiments and analysis on ADOPD, demonstrating its effectiveness and generalization capabilities for document understanding research.

---

[2]The symbol 💻 indicates automatic annotations, 👥 represents human annotations, and 🤖 signifies LLM assistance. 📄 indicates that the document source is a digital PDF, while 🖼 indicates document images.

## 2 RELATED WORK

**Document Datasets.** As shown in Table 1, several recent document image datasets have been introduced. PubLayNet (Zhong et al., 2019) comprises images and annotations generated through the automated alignment of PDFs with XML formats. DocBank (Li et al., 2020b) is created using LaTeX-generated PDF files and employs an efficient weakly supervised approach for annotation. DocLayNet (Pfitzmann et al., 2022) relies on human annotation rather than automated methods. This dataset encompasses six distinct document types and encompasses a total of 11 annotation categories. M⁶Doc (Cheng et al., 2023) is a recently introduced dataset featuring approximately 9k modern document images, divided into seven subsets. It contains detailed annotations spanning multiple distinct categories. IIIT-AR-13K (Mondal et al., 2020) is tailored for object detection in business documents like annual reports, containing annotated pages with standard layout elements like text, headings, lists, graphics, and tables. In summary, existing large-scale document image datasets mainly focus on PDFs, unlike the varied scanned or photographed images encountered in real-world scenarios. This limited distribution of data can bias trained models. Additionally, publicly available datasets often cover only a narrow range of document layouts and categories.

**Document Models.** The document domain has witnessed the emergence of foundational models (Li et al., 2020a; Prasad et al., 2020), driven by advancements in deep learning. Despite rapid progress in document understanding models, the scarcity of powerful models trained on high-quality, large-scale document data remains a significant challenge. Earlier document layout analysis methods (Ouwayed & Belaïd, 2012; Lee et al., 2019) relied heavily on rule-based and heuristic algorithms. However, their applicability was limited to simple document types, resulting in poor generalization performance. In addition to task-driven models, researchers have proposed a range of document pretraining models (Huang et al., 2022; Li et al., 2021; Gu et al., 2021; Tang et al., 2023; Kim et al., 2022). These models are typically pretrained on the IIT-CDIP (Lewis et al., 2006) dataset and evaluated on various document benchmarks. Despite the remarkable performance of these models on benchmark datasets, it is critical to acknowledge that most current image-based document datasets are predominantly composed of a narrow range of document types, failing to capture the heterogeneity of real-world documents. Moreover, the restricted data diversity in these benchmark datasets constrains the development and evaluation of document models.

## 3 ADOPD DATASET

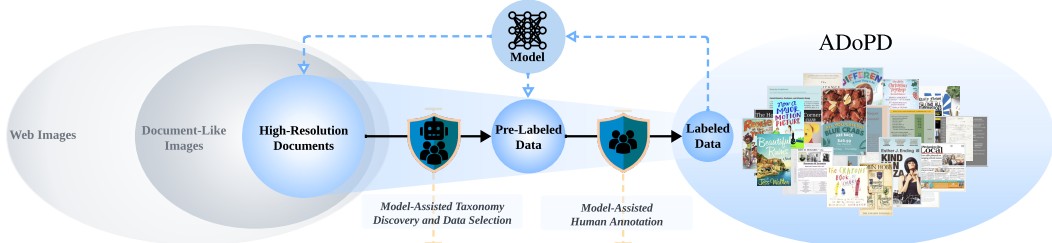

Figure 3: Model-assisted data collection and annotation pipeline for ADOPD.

ADOPD stands out among document datasets as it is constructed using diverse document images found on the web. Sec. 3.1 introduces document page decomposition tasks. Sec. 3.2 presents a data-driven approach to discovering document taxonomy for data collection and analysis. Sec. 3.3 employs models to assist with human annotation, addressing challenges posed by diverse data.

### 3.1 TASK DEFINITION

Fig. 2 illustrates the document page decomposition task defined in this paper, which encompasses four subtasks: Doc2Mask, Doc2Box, Doc2Seq, and Doc2Tag.

- The Doc2Mask task entails segmenting visual entities in document images in a class-agnostic manner. The "entity" in this context denotes a thing (instance) mask or a stuff mask. For example, in Fig. 1, an entity represents a meaningful and coherent region (*e.g.*, banner, figure, logo, *etc*).
- The Doc2Box task calls for identifying text region-of-Interest (RoI) within a document image, regardless of their specific types. The term "box" refers to text RoI (*e.g.*, paragraphs and titles, *etc*).

- The Doc2Seq task involves generating captions for document images, requiring the model to analyze visual elements and structured text. Given the complexity of document images, the model must effectively comprehend visual, textual, and layout information to produce detailed captions.
- The Doc2Tag task is akin to image tagging, specifically multi-label image recognition, where the objective is to assign multiple semantic labels to an image. In Doc2Tag, two levels of tagging are utilized: one based on the overall image content and another on specific local regions.

## 3.2 DATA-DRIVEN DOCUMENT TAXONOMY DISCOVERY

In standard classification scenario, we deal with a given dataset denoted as $\mathcal{D}_{full}$, where $\mathcal{X}$ represents the input space, and $\mathcal{Y} = \{1, \ldots, K\}$ is the label space. The classification model, denoted as $f := g \circ h$, consists of a feature extractor $h : \mathcal{X} \to \mathbb{R}^d$ and a classifier $g : \mathbb{R}^d \to \mathbb{R}^K$, which maps the input's feature embedding to $K$ real-valued numbers called logits. In practice, establishing a guiding taxonomy associated with $K$ is crucial for effective data collection, enabling us to manage and assess the diversity of the collected data. However, determining an appropriate value for $K$ in documents is challenging due to the diversity of documents. We draw inspiration from pretrained models such as CLIP, GPT-4, *etc*, which have been trained on large-scale datasets and can serve as knowledgeable "experts" for data selection. Despite the benefits of pretrained models, the predictions from such models are not always reliable. *E.g.*, LLMs[3] tend to suffer from hallucination problems (Bang et al., 2023). Hence, incorporating safeguards into data collection is essential. Fig. 3 provides an overview of our data collection process, which will be detailed in the subsequent sections.

**Can Large-Scale pretrained Models Facilitate Data Collection?** Given a document image $x \sim \mathcal{D}_{full}$, we can extract document information using pre-existing models as follows:

$$\{z, S_{OCR}, S_{Caption}, S_{Attribute}, S_{Label}\} = \{h(x), f_{OCR}(x), f_{I2T}(x), f_{Tag}(x), f_{CLIP}(x|\mathcal{Y})\} \quad (1)$$

$$\{S^*_{Caption}, S^*_{Tag}\} = \text{LLM}( S_{OCR}, S_{Caption}, S_{Attribute}, S_{Label}|\text{Prompt}) \quad (2)$$

where $z \in \mathbb{R}^D$ is obtained through an image feature extractor $h(\cdot)$. The sequence $S_{OCR}$ consists of words and their coordinates, extracted by OCR tool $f_{OCR}(\cdot)$. The caption $S_{Caption}$ is generated by the captioning model $f_{I2T}(\cdot)$. Tags $S_{Attribute}$ are produced by the image tagging model $f_{Tag}(\cdot)$. Labels $S_{Label}$ are generated by the CLIP model $f_{CLIP}(\cdot|\mathcal{Y})$, constrained by $\mathcal{Y}$. Integrating multimodal information, as expressed in Eq.1, for document reasoning poses a significant challenge. As demonstrated in Eq.2, we harness the power of LLMs and formulate prompts to predict tags ($S^*_{Tag}$) and captions ($S^*_{Caption}$) for document images. The ablation study of these prompts is explored in the Appendix.

**How to Safeguard Data Collection?** Despite the impressive zero-shot capabilities of LLMs for sequence reasoning, prediction errors and uncertainties may still arise. Some failure cases can be addressed with stricter prompts. Even so, fully relying on LLMs for data selection poses heavy risks.

Fig. 4 illustrates our data selection diagram, strengthened by outlier detection. For each batch of sampled web images ($\mathcal{D}_{selected}$), we define it as a mix of in-distribution (ID) ($\mathcal{D}_{pseudo\text{-}in}$) and OOD ($\mathcal{D}_{pseudo\text{-}out}$) data. In $\mathcal{D}_{pseudo\text{-}in}$, all samples belong to taxonomy classes we have already explored, while $\mathcal{D}_{pseudo\text{-}out}$ comprises samples from document types we haven't explored yet. Alg. 1 outlines the process where we integrate outlier detection for data collection and taxonomy discovery. Given the dataset pool denoted as $\mathcal{D}^t_{full}$ and $t$ indicates the time step, we initially select a batch of data, denote as $\mathcal{D}^t_{selected\text{-}in}$, from

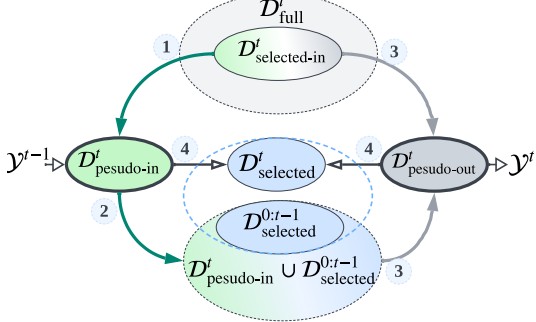

Figure 4: Data selection diagram for ADOPD.

$\mathcal{D}_{full}$. Based on the current taxonomy $\mathcal{Y}^t$, we first partition $\mathcal{Y}^t$ into 100 clusters using the $K$-means algorithm (Jiang et al., 2024). Afterwards, we sample $\mathcal{D}^t_{pseudo\text{-}in}$ from $\mathcal{D}^t_{selected\text{-}in}$ based on the $K$ clusters, corresponding to step ① in Fig. 4. Specifically, we randomly select a sub-category $y^t_k$ from cluster $k$ as the representative category. We then use CLIP to sample $n^t$ documents classified under $y^t_k$. The pseudo input ID data $\mathcal{D}^t_{pseudo\text{-}in}$ comprises a total of $100 \cdot n^t$ document images. This selection process ensures a balanced sampling operation within the current taxonomy $\mathcal{Y}^t$.

---

[3]Without specific indication, LLM in this paper refers to GPT-4.

Sampling from ID data can lead to biased distributions because models trained on such data may silently fail when faced with OOD inputs. Therefore, enhancing the diversity of ADOPD by incorporating hard negative examples may result in an overall improvement in diversity. Therefore, we explicitly sample a OOD subset $\mathcal{D}^t_{\text{pseudo-out}}$ from the current candidate pool $\mathcal{D}^t_{\text{selected-in}} \setminus \mathcal{D}^t_{\text{pseudo-in}}$, corresponding to step ③ in Fig. 4. To obtain $\mathcal{D}^t_{\text{pseudo-out}}$, we employ $K$-means to segregate outliers from $\mathcal{D}^t_{\text{selected-in}}$. Specifically, we extract image features $\bar{\mathbf{Z}}^t$ for $\mathcal{D}^{0:t-1}_{\text{selected}} \cup \mathcal{D}^t_{\text{selectd-in}}$, where $\mathcal{D}^{0:t-1}_{\text{selected}} = \mathcal{D}^{0:t-1}_{\text{pseudo-in}} \cup \mathcal{D}^{0:t-1}_{\text{pseudo-out}}$. In this context, $\bar{\mathbf{Z}}^t = \bar{z}^t_k$, with $k \in [0, 100)$ representing the set of K-means centroids estimated from $\mathcal{D}^{0:t-1}_{\text{selected}} \cup \mathcal{D}^t_{\text{pseudo-in}}$. The outlier score is estimated as the Euclidean distance between it and the nearest centroid:

$$s^t(\boldsymbol{z}) = \min_{k \in [0,100)} ||\boldsymbol{z} - \bar{\boldsymbol{z}}^t_k||_2, \quad \boldsymbol{z} \in (\mathcal{D}^t_{\text{selected-in}} \setminus \mathcal{D}^t_{\text{pesudo-in}}) \tag{3}$$

where $\mathcal{D}^t_{\text{pseudo-out}}$ contains data points with outlier scores ranked in the top $n^t$ across $K$ clusters.

Given the selected ID and OOD data, we have $\mathcal{D}^t_{\text{selected}} = \mathcal{D}^t_{\text{pseudo-in}} \cup \mathcal{D}^t_{\text{pseudo-out}}$, which is ready for annotation (step ④ in Fig. 4). Before annotation, we update $\mathcal{Y}^{t-1}$ by using the newly selected data $\mathcal{D}^t_{\text{selected}}$. Here, we employ the approach outlined in Eq. 2, leveraging the LLM to predict the presence of new labels and obtain the updated taxonomy $\mathcal{Y}^t$. We use prompt-based methods to predict document tags by considering four aspects: *visual* ($\mathscr{P}_{\text{visual}}$), *textual* ($\mathscr{P}_{\text{textual}}$), *layout* ($\mathscr{P}_{\text{layout}}$), and *multimodal* ($\mathscr{P}_{\text{multimodal}}$). Each aspect is addressed through unique input combinations in the prompt. Additional details about the prompts can be found in the Appendix. After obtaining outputs, we implement two safeguards to filter out failures. Firstly, we design a prompt-based summarizer

---

**Algorithm 1:** Data-Driven Taxonomy Discovery

**Input** : $\mathcal{Y}^0, \mathcal{D}_{\text{full}}, \epsilon$.
**Output:** Expanded Taxonomy $\mathcal{Y}$
**while** *True* **do**
  ① Collect $\mathcal{D}^t_{\text{select-in}}$ from $\mathcal{D}^t_{\text{full}}$ ;
  ② Select $\mathcal{D}^t_{\text{pseudo-in}}$ from $\mathcal{D}^t_{\text{selected-in}}$ based on $\mathcal{Y}^{t-1}$ ;
  ③ Generate image embeddings $\mathbf{Z}$ for $\mathcal{D}^{0:t-1}_{\text{selected}} \cup \mathcal{D}^t_{\text{pesudo-in}}$;
  ③ Calculate $s^t(\boldsymbol{z}), \forall \boldsymbol{z} \in (\mathcal{D}^t_{\text{selected-in}} \setminus \mathcal{D}^t_{\text{pesudo-in}})$ ;
  ③ Select outlier data $\mathcal{D}^t_{\text{pesudo-out}}$;
  **foreach** $x \sim \mathcal{D}^t_{\text{pesudo-in}} \cup \mathcal{D}^t_{\text{pesudo-out}}$ **do**
    ④ Predict new labels using four prompters;
    ④ Update $\mathcal{Y}^t$ with the newly predicted labels;
  ④ Refine $\mathcal{Y}^t$ with human annotator;
  **if** $|\mathcal{Y}^t| > \epsilon$ **then**
    Stop;
  **else**
    $t \leftarrow t + 1$ ;

---

($\mathscr{P}_{\text{summary}}$) using LLM to obtain 10 tags by summarizing the tags predicted through the four prompt strategies. Secondly, after the label generation by LLM, human annotators review and eliminate labels that are confusing or irrelevant to the document.

## 3.3 MODEL-ASSISTED DATA ANNOTATION

**Data Collection.** The images in ADOPD are sourced from the Laion-HR (Laion High Resolution), which comprises high-resolution web images, including multilingual document images. Laion-HR provides a foundation for our multi-lingual multi-modal ADOPD. We leverage pretrained models with humans in the loop to collect and filter data. The process includes the following steps:

- *Model-Assisted Data Selection*: We first select images based on Laion-HR's metadata by applying criteria such as pwatermark < 0.8 and punsafe < 0.5. Then, we construct a document discovery dataset using natural image datasets (*e.g.*, ImageNet, *etc*) and document datasets (*e.g.*, DocLayNet, *etc*). We then finetune a DiT-based binary image classifier (Li et al., 2022a) to identify potential documents (probability > 0.8). Subsequently, we apply an OCR tool (Du et al., 2021), and retain those with a word count exceeding 10. Although metadata provides predictions for watermarks, in order to improve accuracy, we additionally train a watermark detection model to filter watermarked images. We compute MD5 hashes and Hamming distances between images to exclude duplicates, even if document images in Laion-HR have different URLs. Fig. 10 in the Appendix shows the percentage of data selection.
- *Human Selection and Sensitive Verification*: Based on our taxonomy[4] obtained by Alg. 1, we adopt pretrained CLIP model for zero-shot tagging. Human annotators then select safe and valid images for all categories. We do not rigidly specify that images must be print-format documents, but instead suggested the annotators to choose those that resemble documents. Annotators are tasked with filtering the dataset for potentially sensitive information.

---

[4]Note that the taxonomy $\mathcal{Y}$ gradually change with the growth of data collection.

**Data Annotation.** The annotation process of ADOPD prioritizes the core principle of *understanding the document's structure and layout*. We avoid imposing overly rigid constraints on annotation[5].

- *Model-Assisted Manual Annotation*: In the early stage, we utilize a pretrained CropFormer (Qi et al., 2023) to generate pseudo entity masks. Annotators follow guidelines to adjust the masks by adding, modifying, or deleting as needed. After annotating a sufficient amount of data, CropFormer is retrained with the new annotations and serves as the seed model for data preprocessing in the subsequent stage. Through this iterative process, our model progressively reduces annotation costs while simultaneously increasing annotation efficiency. Fig. 12 in the Appendix illustrates the effectiveness of model-assisted annotation. During the annotation process, we provide document captions ($S^*_{\text{Caption}}$) and tags ($S^*_{\text{Tag}}$) to aid annotators in understanding the document.
- *Multi-Task and Multi-Lingual Annotation*: ADOPD stands out from other document datasets for its multi-task and multi-lingual characteristics. Our primary focus is on English and CJK (Chinese, Japanese, Korean) documents, with 60k document images in English and the remaining in the other languages. We reserve a private test set for the competition. Each dataset has four tasks introduced in Sec. 3.1. Specifically, for Doc2Mask annotation, we refrain from imposing semantic constraints on labeling entities, therefore encouraging annotators to come up with open-ended names or descriptions that are accurate (*e.g.*, "*doc-in-doc*", "*banner*", "*infographic*", "*natural image*", *etc*). As our task focuses on document entity segmentation, we do not incorporate label information in segmentation evaluations. For Doc2Box, we have stricter rules which require annotators to comprehend words and group them according to their semantic meaning. The annotation files follow the MSCOCO annotation format.

## 4 EXPERIMENTS

### 4.1 IMPLEMENTATION DETAILS

**Baseline Models.** We experiment on the subset of ADOPD, with training and validation sets comprising 50k and 10k images, respectively. **(1)** Doc2Mask: we evaluate two frameworks: Mask2Former (Cheng et al., 2021) and CropFormer (Qi et al., 2023), to identify which is best stuited for the document page decomposition task. We perform ablation studies on these frameworks using different backbones, such as Swin Transformer (Swin) (Liu et al., 2021), Hornet (Rao et al., 2022), and ViT (Parmar et al., 2018). **(2)** Doc2Box: we similarly benchmark three models: Faster R-CNN (Ren et al., 2015), Deformable-DETR (Zhu et al., 2021), and Cascade Mask-RCNN (MR-CNN) (Cai & Vasconcelos, 2019). We also enhance Cascade Mask-RCNN by incorporating pretrained ViT backbones, specifically DINOv1 (Caron et al., 2021) and Dinov2 (Oquab et al., 2023) with ViT-Adapter (Chen et al., 2022). **(3)** Doc2Seq: we build an encoder-decoder model using pretrained ViT and GPT-2 (Radford et al., 2019), fine-tuned on 80k image-caption pairs for training and 20k for validation. The captions are generated using prompts specified in Eq. 2. Acknowledging the gap between LLM-generated and human annotations, we collect an extra 5k human-annotated validation set for further comparison. **(4)** Doc2Tag: we validate our taxonomy discovery using the CLIP ViT-G/14 model and report the OOD performance on RVL-CDIP (Harley et al., 2015).

We build Doc2Mask using the Detectron2 (Wu et al., 2019) and Doc2Box with MMDetection (Chen et al., 2019). All experiments are run on NVIDIA A100-80GB GPUs. Following standard practices(Ghiasi et al., 2021), we employ an input resolution of 1024×1024, achieved by re-scaling and padding the shorter side of the image. Doc2Mask (CropFormer and Mask2Former) and Doc2Box (Faster R-CNN, Cascade Mask-RCNN) are trained for 15 epochs with a batch size of 32 on 8 GPUs to achieve full convergence. We train Deformable-DETR for 30 epochs due to slow convergence issues. We build other models (Doc2Seq and Doc2Tag) with Huggingface Transformers framework (Wolf et al., 2020). For Doc2Seq, we train it for 50 epochs on 8 GPUs with a total batch size of 800. Finetuning CLIP ViT-G/14 on Doc2Seq data takes 100 epochs on 8x8 GPUs.

**Evaluation Metrics.** We evaluate Doc2Mask and Doc2Box with the mean average recall (mAR) and mean average precision (mAP) metrics. This assessment considers ten overlap thresholds ranging from 0.5 to 0.95 in increments of 0.05 (mAP@0.5-0.95). For OOD evaluation, we use metrics including the Area Under the Receiver Operating Characteristic (AUROC), False Positive Rate at 95% Recall (FPR95), maximum concept matching (MCM) score (Ming et al., 2022), and accuracy

---

[5]This research's data collection and annotation were completed in October 2023.

(ACC). For Doc2Seq, we use the BLEU@$n$ (B@$n$) (Papineni et al., 2002), CIDEr (C) (Vedantam et al., 2015), METEOR (M) (Denkowski & Lavie, 2014) and ROUGE (R) (Lin, 2004) for evaluation.

## 4.2 DOCUMENT PAGE DECOMPOSITION TASKS ANALYSIS

**Comparing the Model Architectures.** Table 2 compares Mask2Former and CropFormer models on Doc2Mask. CropFormer outperforms Mask2Former with similar backbones and pretrained datasets. CropFormer's superiority stems from its integration of image crops alongside full image input, enhancing mask prediction with detailed information. This highlights the model's ability to handle multi-view and local image information, especially in the context of document images.

We compare various object detection models in Table 3, including Faster R-CNN, Deformable-DETR, and Cascade MR-CNN. While Deformable-DETR improves, it doesn't outperform anchor-based detectors like Faster R-CNN and Cascade MR-CNN significantly. Despite achieving a higher mAR, the limited mAP improvement may be due to the distinct data distribution of text boxes, differing from general objects in natural images with clear classification boundaries. Meanwhile, the Cascade MR-CNN, combining Mask R-CNN and Cascade R-CNN, achieves the highest mAP. It enhances instance segmentation performance and aids text detection, especially for words requiring pixel-level feature representation.

Table 2: Results on Doc2Mask with varied backbones (T, B, L) denoting tiny, base, and large types.

| | Backbone | Pretrain | Mask Quality | | | |
|---|---|---|---|---|---|---|
| | | | mAP | AP$_{50}$ | AP$_{75}$ | mAR |
| Mask2Former | Swin$_T$ | EntitySeg | 31.80 | 37.16 | 32.33 | 34.0 |
| | | ImageNet | 28.95 | 34.36 | 29.51 | 31.0 |
| | Swin$_L$ | EntitySeg | 32.81 | 38.14 | 33.17 | 35.3 |
| | | ImageNet | 30.21 | 36.30 | 31.18 | 32.5 |
| | Hornet$_L$ | EntitySeg | 34.39 | 40.09 | 34.95 | 36.9 |
| | | ImageNet | 32.96 | 38.22 | 33.30 | 35.2 |
| | ViT$_B$ | SA-1B | 35.59 | 41.05 | 36.35 | 37.6 |
| | ViT$_L$ | | 35.81 | 40.27 | 36.53 | 37.8 |
| CropFormer | Swin$_T$ | EntitySeg | 35.46 | 41.56 | 35.60 | 38.5 |
| | | ImageNet | 37.71 | 45.30 | 38.20 | 41.0 |
| | Swin$_L$ | EntitySeg | 36.03 | 42.30 | 36.23 | 39.3 |
| | | ImageNet | 37.73 | 44.62 | 38.49 | 40.7 |
| | Hornet$_L$ | EntitySeg | 35.03 | 40.00 | 35.73 | 37.6 |
| | | ImageNet | 36.06 | 41.84 | 36.69 | 38.7 |
| | ViT$_B$ | SA-1B | 35.87 | 41.92 | 36.73 | 38.4 |
| | ViT$_L$ | | 39.56 | 45.72 | 40.33 | 42.4 |

**Comparing Backbones and Pretraining.** Table 2 also investigates the impact of vision backbone pretrained on various datasets. References to EntitySeg, ImageNet, and SA-1B indicate pretraining on the respective datasets. SAM (Kirillov et al., 2023) pretrained on SA-1B outperforms the Swin/Hornet models trained on ImageNet or EntitySeg. This can be attributed to two factors: firstly, SA-1B is sufficiently large (around 1 Billion). Secondly, while Swin/Hornet architectures are well-suited for segmentation, SAM is trained with pixel-level supervised learning, enabling it to acquire improved pixel-level representations crucial for document image understanding.

Table 3 compares different backbones on Doc2Box. Dinov2$_{P14}$ + ViT$_{Adapter}$ excels with higher mAP yet slightly lower mAR, demonstrating the superiority of self-supervised backbones over pretrained alternatives. This is crucial for document analysis, given the absence of high-quality ImageNet-like pretraining data. Comparing Dinov1$_{P8}$ and Dinov1$_{P16}$ suggests that fine-grained patches enhance document image features. Fig. 5b illustrates the results of Doc2Box using Dinov2$_{P14}$+ViT$_{Adapter}$.

Table 3: Performance comparisons on Doc2Box.

| Method | Backbone | Box Quality | | | |
|---|---|---|---|---|---|
| | | mAP | AP$_{50}$ | AP$_{75}$ | mAR |
| Faster R-CNN | ResNet$_{50}$ | 61.1 | 78.9 | 67.0 | 74.9 |
| | ResNet$_{101}$ | 61.4 | 78.6 | 67.3 | 74.3 |
| Deformable-DETR | ResNet$_{50}$ | 65.0 | 82.2 | 72.1 | 81.6 |
| | ResNet$_{101}$ | 65.5 | 82.8 | 72.7 | 81.6 |
| Cascade MR-CNN | ResNet$_{50}$ | 64.7 | 80.9 | 71.0 | 79.4 |
| | ResNet$_{101}$ | 65.3 | 71.7 | 68.7 | 79.1 |
| | Dinov1$_{P8}$+ ViT$_{Adapter}$ | 63.6 | 80.4 | 69.6 | 76.3 |
| | Dinov1$_{P16}$+ ViT$_{Adapter}$ | 63.2 | 80.3 | 69.5 | 76.2 |
| | Dinov2$_{P14}$+ ViT$_{Adapter}$ | 67.0 | 82.7 | 73.2 | 77.8 |

**Evaluating Generalization Ability.** In Table 4 (a), we compare the model trained on ADOPD with those fine-tuned on EntitySeg. Combined with Fig. 5a, it is evident that models fine-tuned on ADOPD can better focus on fine-grained document elements and make more reasonable predictions for document entity masks. Conversely, models pretrained on EntitySeg can predict some masks but tend to excessively detect elements present in natural images (*e.g.*, people, objects), while neglecting the document's inherent layout. Table 4 (b) validates the cross-dataset generalization pretraining advantage of ADOPD, specifically focusing on the evaluation set of DocLayNet. For a fair comparison, we consider only text detection without categorizing the boxes. Directly applying the model fine-tuned on ADOPD to DocLayNet data yields zero-shot results with high recalls. Furthermore, fine-tuning on DocLayNet with ADOPD pretrained backbones outperforms fine-tuning with ImageNet backbones. Note that DocLayNet's testing is limited to its limited document types and cannot reveal anything about the generalization capability of ADOPD for other taxonomy types.

Table 4: Ablation studies: (a) comparing the performance of models trained with ADOPD and models fine-tuned only on EntitySeg. The results in "(·)" represent the zero-shot outcomes for EntitySeg. (b) Cross-dataset evaluation to assess the generalizability of ADOPD on DocLayNet.

(a) Results for with and without ADOPD.

| Backbone | mAP | $AP_{50}$ | $AP_{75}$ | mAR |
|---|---|---|---|---|
| Mask2Former | | | | |
| $Swin_L$ | 32.81(16.27) | 38.14(22.52) | 33.17(15.88) | 35.3(29.3) |
| $Hornet_L$ | 34.39(15.83) | 40.09(21.74) | 34.95(15.38) | 36.9(29.0) |
| Cropformer | | | | |
| $Swin_L$ | 36.03(13.46) | 42.30(19.41) | 36.23(13.12) | 39.3(24.5) |
| $Hornet_L$ | 35.03(15.68) | 40.00(21.39) | 35.73(15.26) | 37.6(26.9) |

(b) Cross-dataset evaluation on DocLayNet.

| Method | Backbone | Zero-Shot ADOPD | | Finetune ImageNet | | Finetune ADOPD | |
|---|---|---|---|---|---|---|---|
| | | mAP | mAR | mAP | mAR | mAP | mAR |
| Faster R-CNN | $ResNet_{50}$ | 0.9 | 58.5 | 43.0 | 55.8 | 44.5 | 60.4 |
| | $ResNet_{101}$ | 1.0 | 56.6 | 46.0 | 58.5 | 47.0 | 60.7 |
| Deformable-DETR | $ResNet_{50}$ | 2.2 | 80.4 | 74.7 | 87.2 | 75.4 | 88.9 |
| | $ResNet_{101}$ | 2.6 | 79.0 | 75.4 | 85.9 | 77.2 | 88.1 |

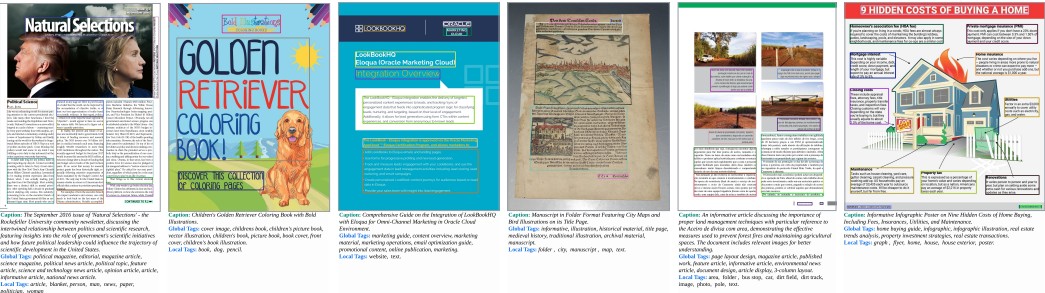

(a) Mask Prediction Comparison: From top to bottom, we showcase the original image and predictions from the best models trained on ADOPD, EntityV2, and SAM (SA1B), respectively.

(b) Document text detection visualization results, each image paired with its caption and tags.

Figure 5: Visualization of ADOPD images and results for Doc2Mask and Doc2Box.

**Prompt-Guided Context-Aware Captioning Benefits Vision-Language Modeling.** Table 4.2 evaluates caption quality. We collect 5K test data to evaluate the effectiveness of Doc2Seq. The 🤖 represents the GPT-4 model, and $BLIP_{Large}$ (Li et al., 2022b) and BLIP2-OPT-2.7b (Li et al., 2023) are obtained from Huggingface model hub. ViTBase-P32-384/ViT$_{Base-P16-384}$+GPT2 are fine-tuned on Doc2Seq. While GPT-4 captions achieve a commendable CIDEr score, indicating consensus, a noticeable disparity persists between them and human annotations. Models fine-tuned on Doc2Seq can attain similar performance to GPT-4 on B@$n$, but show significantly lower CIDEr scores. In Fig. 6 (left), human-written captions are notably longer than machine-generated ones, impacting reference-based evaluation. These findings highlight the challenge in document captioning due to diverse interpretations and varying caption lengths, complicating evaluation.

To verify the benefits of prompt-guided captions, we fine-tune CLIP with Doc2Seq data and conduct two experiments: zero-shot evaluation on RVL-CDIP test set and supervised training on RVL-CDIP based on finetuned CLIP vision backbone. While finetuned CLIP improves zero-shot capability for specific data (*e.g.*, Budget and Presentation), the overall enhancement is comparable to raw pretrained CLIP. We observe from Fig. 6 (right) and Table 6 that finetuning the CLIP ViT backbone, initially trained on Doc2Seq, with a classifier layer for separate training on RVL-CDIP results in a noticeable improvement. This underscores the importance of caption rewriting for handling noisy data.

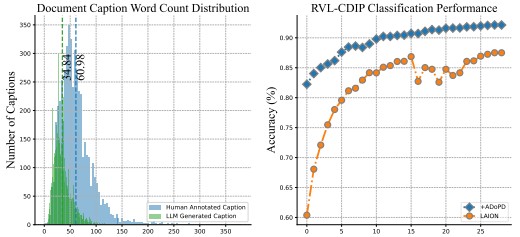

Figure 6: Ablation study on captions.

Table 5: Ablation experiments on Doc2Seq.

| Method | Test | B@1 | B@2 | B@3 | B@4 | M | R | C |
|---|---|---|---|---|---|---|---|---|
| 🤖 | 👥 | 27.0 | 16.7 | 11.7 | 8.5 | 12.8 | 22.4 | 84.7 |
| BLIP$_{Large}$ | 🤖 | 12.4 | 8.5 | 6.4 | 5.0 | 9.3 | 22.6 | 18.3 |
| | 👥 | 8.2 | 5.6 | 4.0 | 3.0 | 8.6 | 21.6 | 3.4 |
| BLIP2-OPT-2.7b | 🤖 | 4.3 | 3.6 | 3.0 | 2.6 | 10.7 | 25.1 | 16.5 |
| | 👥 | 12.3 | 7.6 | 5.3 | 3.9 | 9.0 | 21.8 | 18.0 |
| ViT$_{Base-P32-384}$+GPT2 | 🤖 | 22.5 | 9.2 | 4.4 | 2.5 | 7.7 | 16.8 | 9.8 |
| | 👥 | 16.7 | 5.8 | 2.3 | 1.0 | 5.8 | 13.9 | 4.4 |
| ViT$_{Base-P16-384}$+GPT2 | 🤖 | 23.4 | 9.7 | 4.7 | 2.7 | 8.0 | 17.2 | 11.0 |
| | 👥 | 17.3 | 6.0 | 2.4 | 1.1 | 6.0 | 14.1 | 5.3 |

Table 6: Performance of Models for Per-Class Classification

| Model | Type | Letter | Form | Email | Hw | Ad | SR | SP | SP | FF | NA | Bgt | Inv | Prsnt | Qnr | Rsm | Memo | Avg |
|---|---|---|---|---|---|---|---|---|---|---|---|---|---|---|---|---|---|---|
| | | | | | | | Per-Class (Recall@5) | | | | | | | | | | | |
| DiT$_{base}$ | Supervised | 98.92 | 98.76 | 99.45 | 98.99 | 99.64 | 97.86 | 99.84 | 99.31 | 99.52 | 99.18 | 99.24 | 98.83 | 99.76 | 99.15 | 99.28 | 99.16 | 99.18 |
| ViT$_{G-14}$ | Zero-Shot | 98.52 | 98.76 | 79.78 | 91.89 | 53.46 | 76.51 | 87.64 | 84.82 | 32.81 | 99.49 | 33.07 | 99.35 | 54.81 | 93.75 | 90.46 | 98.23 | 79.58 |
| ViT$_{G-14}$+ADOPD | Zero-Shot | 94.32 | 87.55 | 75.33 | 85.79 | 27.46 | 80.33 | 96.21 | 63.46 | 38.72 | 98.79 | 65.81 | 93.93 | 72.26 | 84.25 | 84.41 | 95.02 | 77.73 |
| | | | | | | | Per-Class (Accuracy) | | | | | | | | | | | |
| DiT$_{base}$ | Supervised | 92.41 | 86.83 | 98.97 | 96.13 | 94.63 | 87.11 | 95.22 | 94.90 | 96.68 | 92.77 | 92.73 | 94.07 | 87.38 | 90.84 | 97.67 | 95.14 | 93.36 |
| ViT$_{G-14}$ | Supervised | 86.20 | 76.70 | 94.28 | 93.48 | 91.81 | 71.94 | 91.10 | 89.72 | 94.97 | 83.68 | 81.24 | 87.44 | 78.26 | 84.97 | 92.94 | 85.87 | 86.57 |
| ViT$_{G-14}$+ADOPD | Supervised | 90.87 | 84.48 | 96.98 | 95.34 | 93.76 | 82.39 | 93.51 | 93.00 | 95.61 | 89.81 | 89.62 | 92.85 | 84.29 | 90.23 | 96.45 | 92.62 | 91.38 |

[1] The abbreviations are: Handwritten (Hw), Advertisement (Ad), Scientific Report (SR), Scientific Publication (SP), Specification (Spec), File Folder (FF), News Article (NA), Budget (Bgt), Invoice (Inv), Presentation (Prsnt), Questionnaire (Qnr), and Resume (Rsm).

**Data-Driven Document Taxonomy Analysis.** To verify Alg. 1, we collect the ID dataset from both RVL-CDIP and Laion-HR based on the 16 classes provided in RVL-CDIP. We sample OOD categories such as "*Magazine (M)*", "*Comic (C)*", "*Guidebook (G)*", "*Yearbook (Y)*", "*Worksheet (W)*", and "*Open Book (OB)*", *etc*, from $\mathcal{Y}$ and collect the OOD data from Laion-HR. In the Appendix, Table 7 shows OOD detection results for two variants: predicting 16 and 50 centroids separately. The K-means method with 50 centroids excels in detecting outliers across all categories. Fig. 7 (center) displays taxonomy expansion with HITL taxonomy cleaning. We start with an initial ID set with 10 classes selected from RVL-CDIP. At every step, we sample 10 detected outlier data. As the data increases, our outlier detection method successfully retrieves outliers for the majority of novel categories. Fig. 7 (left, right) illustrates the distribution of "*Comic*" and ID data, where "*Comic*" is detected as an outlier in the first step. Red color indicates the detected outlier samples.

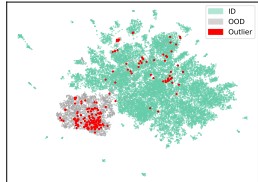 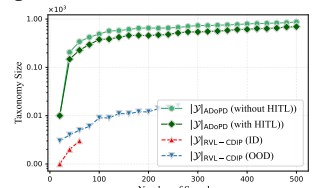 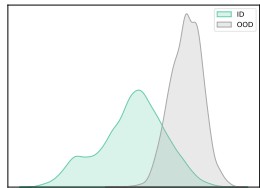

Figure 7: Visualization of outlier detection, taxonomy expansion, and OOD score distributions.

**Responsible AI Analysis.** During data cleaning, we conduct a comprehensive Responsible AI analysis, tackling biases in sensitive areas such as nudity, sexuality, and violence, *etc*. We meticulously filter sensitive data with input from 15 diverse evaluators. Fig. 8 displays their geographic distribution. If any evaluator deems an

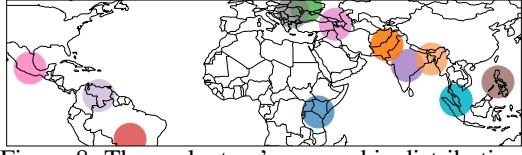

Figure 8: The evaluators' geographic distribution.

image inappropriate, we label it as sensitive. After review, we remove 9.29% of potentially sensitive images, ensuring the majority of the 120K images remain non-sensitive. This rigorous process guarantees a safer and less biased dataset, promoting fairness and inclusivity in our models.

# 5 CONCLUSION

This paper introduces ADOPD, a large-scale dataset for document page decomposition, and outlines a systematic process including data collection, taxonomy analysis, model-assisted data annotation, and HITL processes. We conduct comprehensive analyses and detailed experimental comparisons across four tasks, demonstrating the value of ADOPD. It opens up numerous opportunities for future exploration and the development of foundational models for document understanding, aiming to catalyze advancements in document analysis.

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

# A APPENDIX

## A.1 ABLATION STUDY OF OUTLIER DETECTION

Table 7 presents OOD detection results obtained using our K-means-based method, with two variants: predicting 16 and 50 centroids separately. The K-means method with 50 centroids excels in OOD detection across all categories, showcasing its effectiveness in identifying outlier data.

Table 7: Performance of outlier detection methods in dataset exploration.

| ID | Outlier Score | AUROC (%)↑ | | | | | | FPR95 (%)↓ | | | | | |
|---|---|---|---|---|---|---|---|---|---|---|---|---|---|
| | | M | C | G | Y | W | OB | M | C | G | Y | W | OB |
| RVL-CDIP | K-Means (16) | 99.99 | 99.99 | 99.99 | 99.99 | 99.85 | 99.98 | 0.01 | 0.11 | 0.00 | 0.00 | 0.67 | 0.08 |
| | K-Means (50) | 99.99 | 99.99 | 99.99 | 99.99 | 99.87 | 99.97 | 0.01 | 0.03 | 0.00 | 0.00 | 0.53 | 0.16 |
| | MCM | 80.23 | 94.50 | 91.36 | 92.44 | 91.92 | 89.57 | 80.58 | 31.27 | 51.01 | 42.11 | 44.06 | 49.37 |
| ADOPD | K-Means (16) | 86.28 | 90.86 | 89.08 | 87.64 | 59.16 | 78.13 | 33.02 | 21.40 | 28.05 | 39.61 | 82.66 | 65.74 |
| | K-Means (50) | 86.89 | 92.86 | 89.41 | 87.46 | 59.90 | 78.71 | 32.35 | 27.69 | 29.53 | 39.85 | 80.46 | 65.70 |
| | MCM | 71.42 | 91.13 | 86.69 | 87.58 | 87.58 | 84.16 | 89.89 | 45.18 | 68.23 | 54.39 | 56.66 | 60.76 |

### A.1.1 ABLATION STUDY OF PROMPTS

Table 8: Human evaluation of prompt-guided context-aware captioning.

| Prompt | 👥 | 👥 | Prompt | 👥 | 👥 |
|---|---|---|---|---|---|
| ① ⚙ You are an expert in generating descriptive captions for documents. Provide a concise reference description of the document: `<image caption>` and some image attributes: `<tags>` . The document has words extracted from the image: `<ocr words>` . 💬 Create a comprehensive and precise caption to describe the document content. | 27 | | ② ⚙ You are an expert in generating descriptive captions for documents. Provide a concise reference description of the document: `<image caption>` and some image attributes: `<tags>` . The document has words extracted from the image: `<ocr words>` . 💬 Create a comprehensive and precise caption to describe the document content. | 38 | |
| ③ ⚙ You are an expert in generating caption for document image. The document includes some visual attributes: `<tags>` , and contains text content: `<ocr words>` . There is a possible coarse caption for the document: `<image caption>` . 💬 Now, in your own words, please describe the document's content, referencing the provided information. | 41 | 61 | ④ ⚙ You are an expert in generating descriptive captions for documents. Provide a concise reference description of the document: `<image caption>` and some attributes: `<tags>` . The document has some text content: `<ocr words>` . Please describe the document's content, referencing the provided information. 💬 Create a comprehensive and precise caption to describe the document content. | 47 | 48 |
| ⑤ ⚙ You are an expert in generating caption for document image. The document includes some visual attributes: `<tags>` , and contains text content: `<ocr-summary>` . There is a possible coarse caption for the document: `<image caption>` . 💬 Now, in your own words, please describe the document's content, referencing the provided information. | | 44 | ⑥ ⚙ You are an expert in generating descriptive captions for documents. Provide a concise reference description of the document: `<image caption>` and some attributes: `<tags>` . The document has some text content: `<ocr-summary>` . Please describe the document's content, referencing the provided information. 💬 Create a comprehensive and precise caption to describe the document content. | | 49 |

Table 8 shows the results of human evaluation for various prompts. In each case, we randomly select 100 images for comparison and instruct human evaluators to exclude any incorrect samples.

As described in Sec. 3.2, we employ four prompts to generate annotated document labels, utilizing textual, visual, spatial, and multimodal information. The detailed prompts are listed in Table 9. While the format of prompts focused on different aspects remains consistent, each prompt includes different input items, as indicated by check signals in Table 9. The bold text in the table represents inputs or variables related to prompt selection, as shown in Table 10.

Considering the impact of input item order on taxonomy generation quality, we conduct experiments on various prompts to explore optimal input sequencing. Our investigation focuses on prompt sentences containing inputs (row 3-7 in Table 9). To simplify, we specifically examine rows 6 and 7, both representing high-level information for entire document images and intended to pair together, while the order between them remains undecided. In this study, human annotators evaluate prompt output labels and vote on them, assessing relevance between the document image and labels, eliminating orders resulting in conflicting or ambiguous labels. Subsequently, we tally votes for each order to determine the most favored arrangement.

1. **Order of Visual and Textual Prompt Inputs:** We shuffle the order of input items for the visual prompt (3, 6, and 7) into four different arrangements while keeping 6 and 7 adjacent: 3-6-7, 3-7-6, 6-7-3, and 7-6-3. The investigation result is depicted in Fig. 9(a). Similar to the visual prompt, for the textual prompt, we investigate four orders: 5-6-7, 5-7-6, 6-7-5, and 7-6-5. The investigation result is shown in Fig. 9(b). Based on the results of the visual and textual Prompts, we select 3-6-7 and 5-6-7 as the input order for our prompt. Additionally, we observe that it's preferable to input the description (row 6) before the pre-labels (row 7) and place it at the end of the list. We will maintain this order in the subsequent study.

2. **Order of Layout Prompt Inputs:** We investigate two different layout orders: 4-5-6-7 and 5-4-6-7. Considering that text space may influence layout comprehension, we classify a document with a text area larger than 0.2 of the size as text-rich document. In our study, we present the results separately based on this criterion in Fig. 9(c). The figure illustrates that when the text is rich, the order 5-4-6-7 receives more votes, while 4-5-6-7 is favored otherwise. Consequently, we choose to use the order 5-4-6-7 when the document is rich and 4-5-6-7 when the document is not as text-rich.

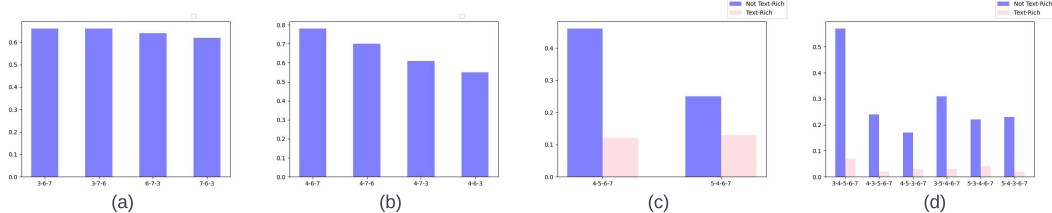

Figure 9: Human evaluation result for different order of Visual and Textual prompt inputs.

Table 9: Prompts across four aspects: textual, visual, layout, and multimodal.

| | | Dataset | Visual | Textual | Layout | Multimodal |
|---|---|---|---|---|---|---|
| | 1 | You are an expert in seeing and understanding a given document image. | ✓ | ✓ | ✓ | ✓ |
| | 2 | Here is the relevant information about the document: | ✓ | ✓ | ✓ | ✓ |
| | 3 | Visual elements may appear in the document: `visual [Tag`$_0$`, Tag`$_1$`, ···]`. | ✓ | ✗ | ✗ | ✓ |
| | 4 | OCR layout information: `[[OCR`$_0$`, Bbox`$_0$`], [OCR`$_1$`, Bbox`$_1$`], ···]`. In each layout item, the first 4 numbers $(x_1, y_1, x_2, y_2)$ are normalized coordinates, and the last string is the detected OCR text | ✗ | ✗ | ✓ | ✓ |
| | 5 | The textual content in the document are: `[OCR`$_0$`, OCR`$_1$`, ···]`. | ✗ | ✓ | ✓ | ✓ |
| | 6 | A possible coarse description of this document: `image caption`. | ✓ | ✓ | ✓ | ✓ |
| | 7 | Possible overall document categories: [Pre-label$_0$, Pre-label$_1$,···]. | ✓ | ✓ | ✓ | ✓ |
| | 8 | You will primarily focus on `USED RESOURCE` to provide 5 labels related to the conception of "CONCEPT" of documents in your own word. The labels should be significantly representative of this document within 5 words. | ✓ | ✓ | ✓ | ✓ |
| | 9 | Here is an example for output: `[Label`$_0$`, Label`$_1$`, ···]`. The labels should not contain any specific content shown in the document. | ✓ | ✓ | ✓ | ✓ |

3. **Order of Mutimodal Prompt Inputs:** Multimodal prompts predict labels using all provided information. Building on previous findings, we standardize the order of rows 6 and 7, placing them at the bottom of the inputs. For the remaining 3 inputs, we shuffle them to create a total of 6 different orders: 3-4-5-6-7, 4-3-5-6-7, 4-5-3-6-7, 3-5-4-6-7, 5-3-4-6-7, and 5-4-3-6-7. As shown in Fig. 9, the results suggest that the order 3-4-5-6-7 yields the best performance among all orders. After obtaining labels with different information, we summarize them into 10 categories to classify the document. The detailed prompt is shown in Table 11.

Table 10: Utilized information on various settings for prompts.

| Modality | Information |
|---|---|
| Visual | The potential visual elements are considered alongside other mentioned details such as descriptions and possible categories. |
| Textual | The textual content is summarized while taking into account other mentioned details such as descriptions and possible categories. |
| Layout | The layout information and design of the document are considered alongside relevant details such as descriptions, visual elements, textual content, and possible categories. |
| Multimodal | All the aforementioned information, including descriptions, visual elements, textual content, possible categories, and layout information, is taken into consideration. |

Table 11: Prompts for summarizing document labels.

| | | Prompt Sentence |
|---|---|---|
| | 1 | You are a languistic expert that can well understand the difference between phrases. |
| | 2 | You will receive 4 lists of labels of the same document image from 4 annotators. They focus on visual, textual, layout, and multimodal aspects of the image. |
| | 3 | You should come out 10 most common conceptions from the list with in 4 words for each. |
| | 4 | The labels given by the visual-focused annotator are: `[Label`$_0^{Visual}$`, Label`$_1^{Visual}$`, ···]`. |
| | 5 | The labels given by the linguistic-focused annotator are: `[Label`$_0^{Textual}$`, Label`$_1^{Textual}$`, ···]`. |
| | 6 | The labels given by the layout-focused annotator are: `[Label`$_0^{Spatial}$`, Label`$_1^{Spatial}$`, ···]`. |
| | 7 | The labels given by the multimdal-focused annotator are: `[Label`$_0^{Multimdal}$`, Label`$_1^{Multimdal}$`, ···]`. |
| | 8 | Here is a example for output: `[Label`$_0$`, Label`$_1$`, ···]`. |

## A.2 DATA ANNOTATION PROCESS ANALYSIS

### A.2.1 IMAGE ANALYSIS

Fig. 11 illustrates our analysis of the document image data, showcasing the high resolution of our dataset. During the annotation process, we notice that despite having high resolution, some document images' text appears blurry. We require annotators to skip labeling such images to ensure clear visibility of text in all document images.

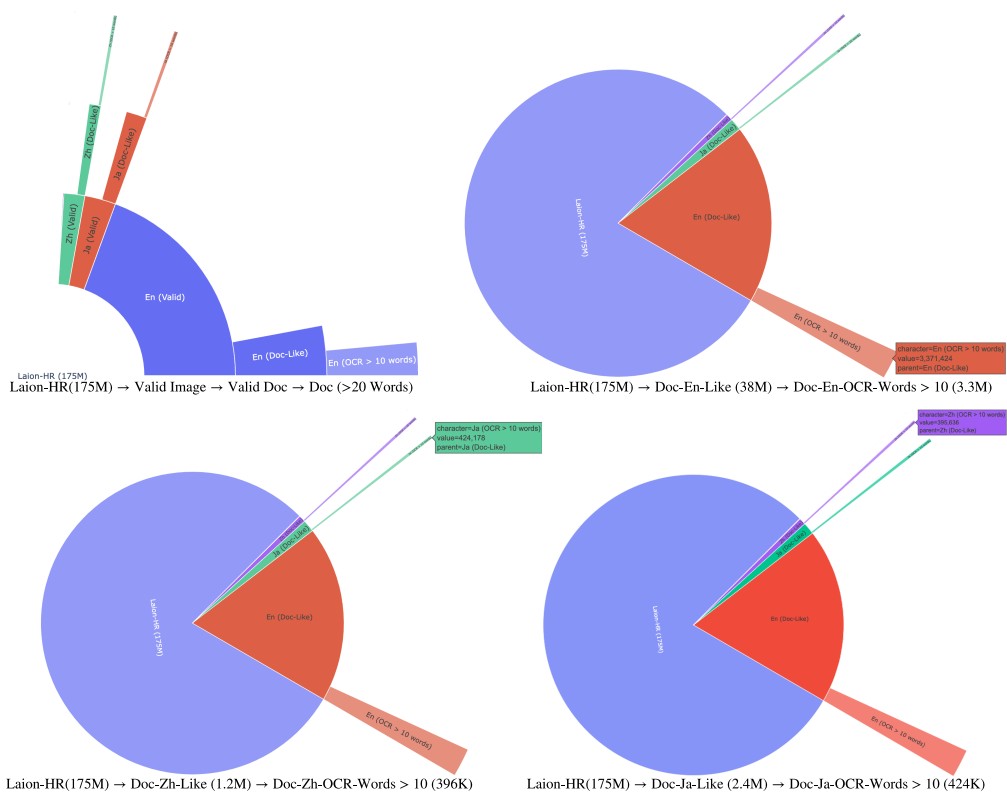

Figure 10: Diagram illustrating data selection proportions.

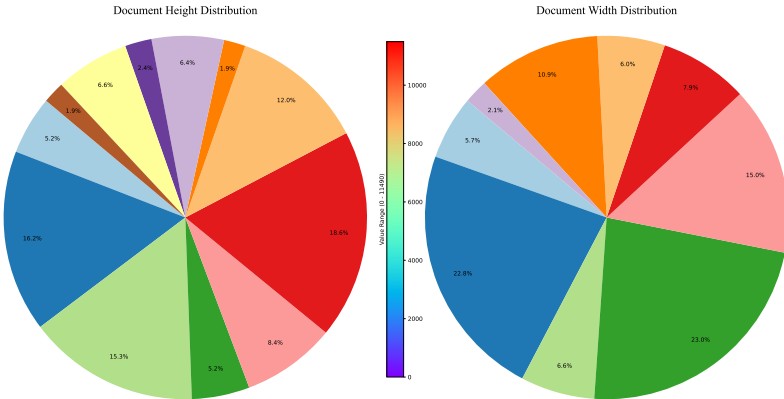

Figure 11: Distribution of the document image sizes for ADOPD.

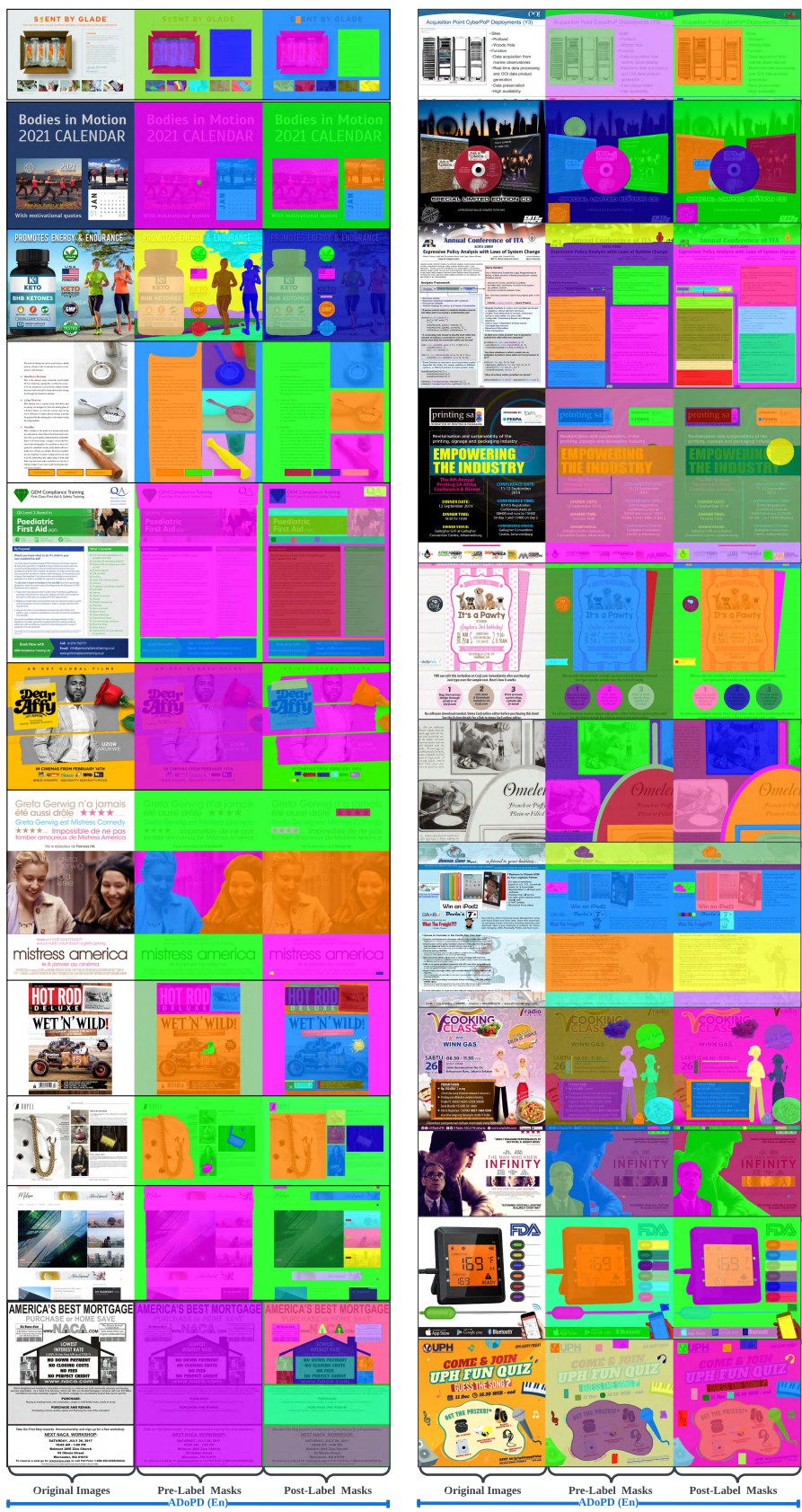

Figure 12: ADOPD (En) model-assisted annotation comparison. (Left) EntitySeg labels (prediction) and human annotations. (right) Doc2Mask (finetune on ADOPD) labels (prediction) and human annotations.

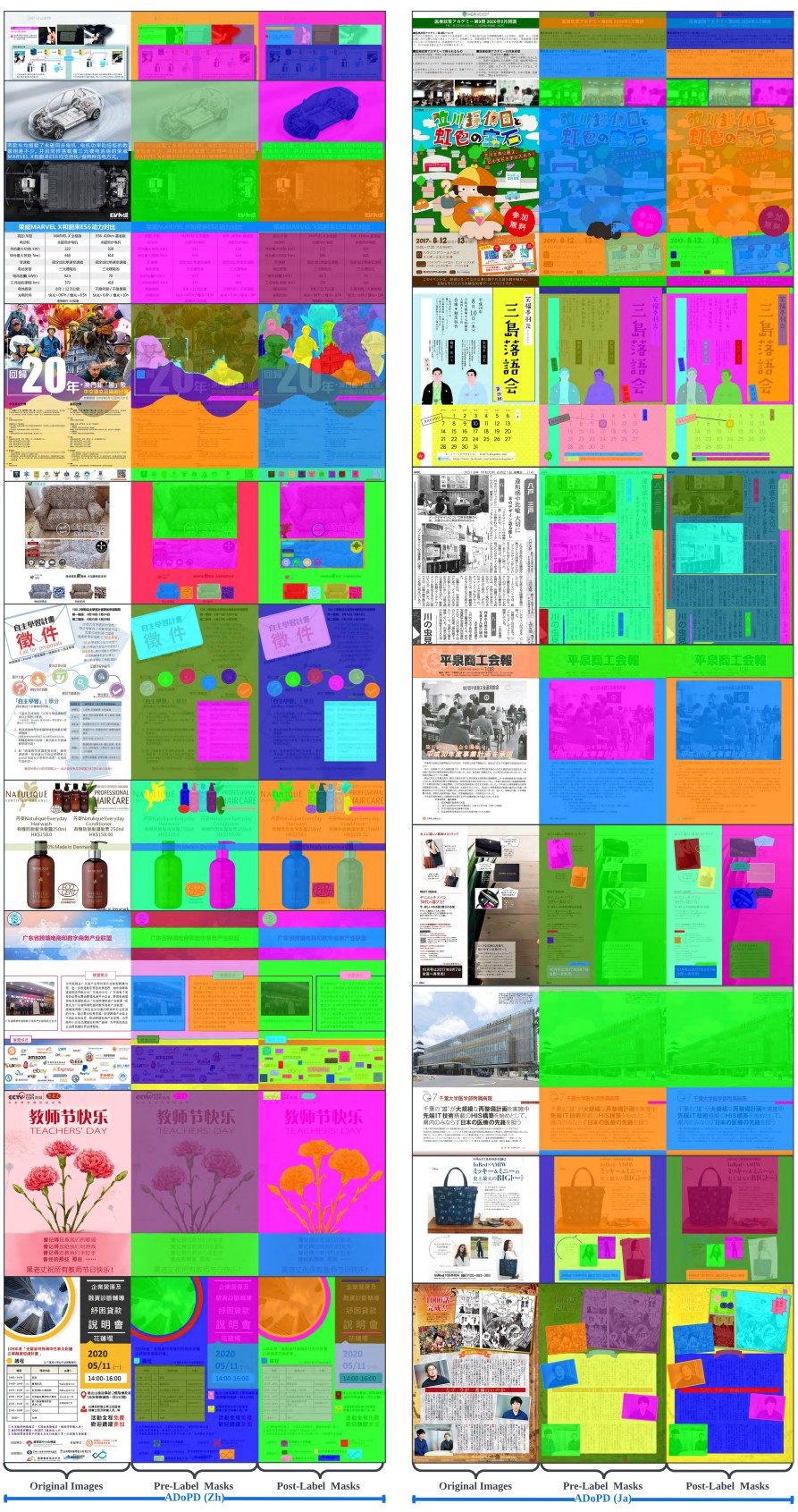

Figure 13: ADOPD (Non-En) model-assisted annotation comparison. (Left) Doc2Mask (finetune on ADOPD-En) labels (prediction, Zh) and human annotations. (Right) Doc2Mask (finetune on ADOPD-En) labels (prediction, Ja) and human annotations.

### A.2.2 MODEL-ASSISTED ANNOTATION DATA VISUALIZATION

### A.3 MULTI-LINGUAL DOC2BOX SAMPLES

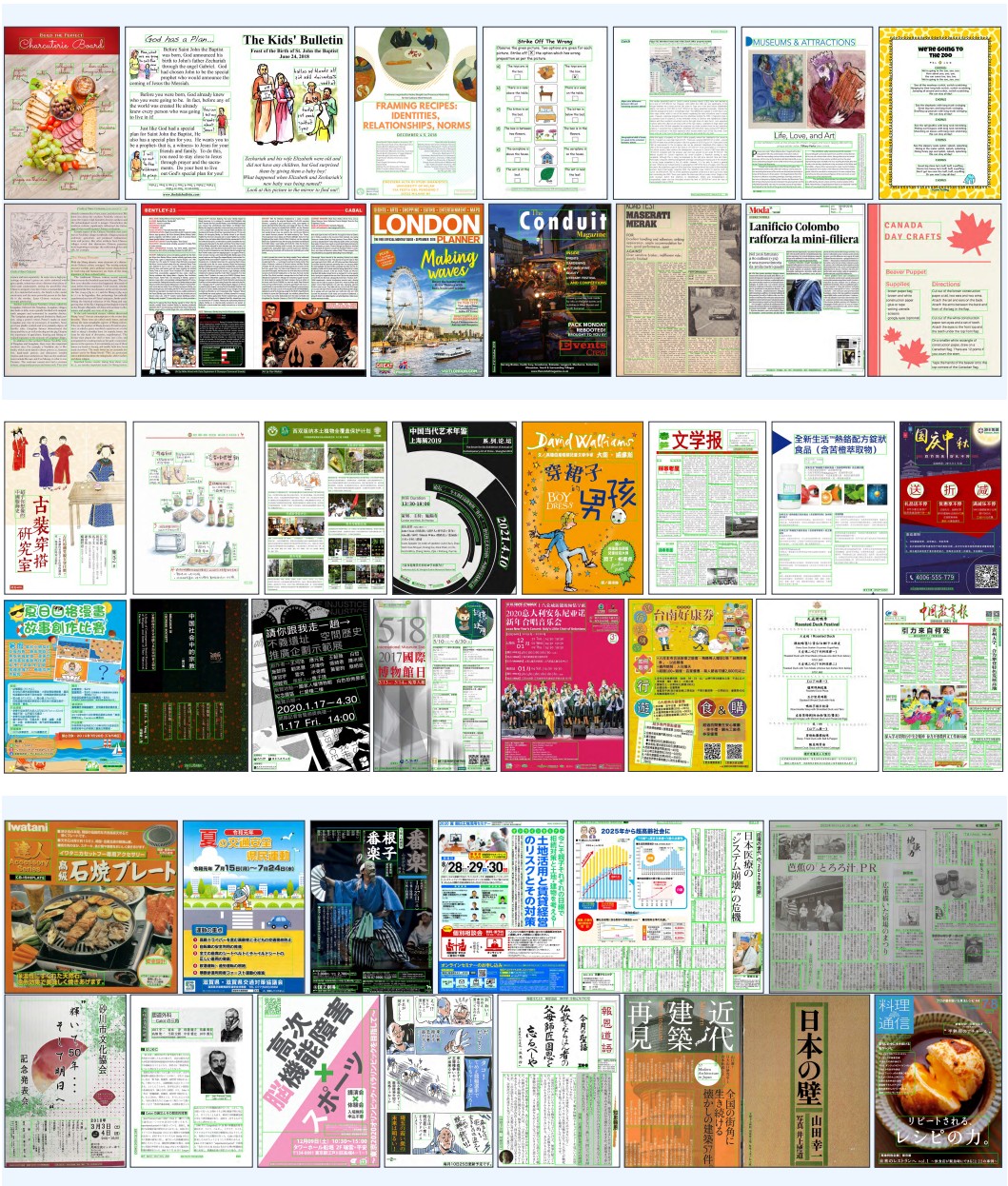

Figure 14: Doc2Box samples: English, Chinese, and Japanese document images, top to bottom. Japanese's unique style poses challenges for Doc2Box compared to English and Chinese.

