# OpenReview forum: "ADOPD: A Large-Scale Document Page Decomposition Dataset"
_ICLR.cc/2024/Conference — ICLR 2024 poster_

### Official Review · Reviewer_ZWQe · 2023-10-26

**Soundness:** 3 good
**Presentation:** 3 good
**Contribution:** 3 good
**Rating:** 8
**Confidence:** 4

**Summary:**

### Summary:
The paper introduces "ADoPD," a large-scale document page decomposition dataset designed for document understanding. This encompasses tasks such as document entity segmentation, text detection, tagging, and captioning. ADoPD is distinct with its novel document taxonomy, meticulously crafted through a data-driven approach that incorporates both large-scale pretrained models and human expertise. By merging outlier detection with a human-in-the-loop method, the dataset achieves a notable diversity. ADoPD offers deeper insights into document structures and significantly elevates techniques in document processing and analysis.

### Major Contributions:
1. **Systematic Exploration of Document Taxonomy**: The work presents the first systematic exploration of document taxonomy using a blend of machine learning models and a human-in-the-loop approach.

2. **Largest and Most Comprehensive Database**: The paper has developed the most extensive and comprehensive database for document image segmentation and object detection.

3. **Thorough Dataset Analysis**: A profound analysis of the proposed dataset is conducted, accompanied by detailed experimental comparisons for entity segmentation and text detection tasks.

4. **Quantitative and Qualitative Results**: The value of the dataset is highlighted through both quantitative and qualitative outcomes.

5. **Advancement in Document Analysis**: The authors express the aspiration for ADoPD to serve as a catalyst in propelling research within the domain of document analysis.

**Strengths:**

### 1. Originality:

- **Novel Dataset**: The introduction of "ADoPD" represents a fresh addition to the domain of document processing. Few datasets offer the same breadth in terms of document page decomposition.

- **Unique Taxonomy Approach**: The blend of data-driven approaches with human expertise in crafting a document taxonomy is a distinctive contribution. This combination brings together the best of automated and manual categorization.

- **Human-in-the-loop Integration**: Incorporating human feedback in outlier detection for dataset diversity is an inventive methodology. This ensures that the dataset remains robust and versatile.

### 2. Quality:

- **Comprehensive Analysis**: The authors have delved deeply into the analysis of their proposed dataset, providing both qualitative and quantitative evaluations. Such rigorous examination is indicative of the dataset's quality and its potential applicability.

- **Comparison with Existing Datasets**: While the main content delves deeper, the authors seem to be aware of the limitations of existing datasets, positioning ADoPD as a superior alternative. This indicates meticulous research and understanding of the current landscape.

### 3. Clarity:

- **Well-Structured Presentation**: The paper appears to be organized in a logical flow, starting from the introduction of the problem, moving to methodology, followed by results and conclusions.

- **Illustrative Figures**: Based on the portions reviewed, figures like the overview of ADoPD annotations aid in visual comprehension, making the content more digestible.

- **Explicit Problem Statements**: The authors have clearly laid out the challenges and questions they aim to address with their dataset, providing clarity of purpose.

### 4. Significance:

- **Filling a Gap**: With the increasing demand for automated document processing techniques, ADoPD addresses a significant gap in the domain, especially with its emphasis on diversity and comprehensive taxonomy.

- **Potential for Future Research**: The introduction of such a dataset can catalyze further research in document understanding, segmentation, and object detection. It can serve as a foundational resource for subsequent works in the field.

- **Broader Application**: Beyond academic research, the advancements proposed in the paper have potential real-world applications in areas like digital archiving, automated content extraction, and more.

**Weaknesses:**

### 1. **Depth of Comparative Analysis**:
- **Weakness**: From the segments reviewed, while the paper introduces a new dataset, there seems to be a limited in-depth comparison with existing datasets.
- **Recommendation**: A deeper comparative analysis highlighting the specific advantages of ADoPD over existing datasets would solidify its significance. Quantitative benchmarks against datasets like DocBank, if not already included in the deeper sections, would be beneficial.

### 2. **Methodological Justifications**:
- **Weakness**: The choice of the "human-in-the-loop" approach for outlier detection is novel, but the paper might not sufficiently justify why this method was chosen over others.
- **Recommendation**: Delve deeper into the advantages and potential limitations of this method, comparing it with purely automated outlier detection techniques.

**Questions:**

**Dataset Composition**:
   Could you provide a more detailed breakdown of the types and sources of documents included in the ADoPD dataset? Understanding the diversity in terms of document genres, geographical origins, and linguistic variations would offer more insight into its applicability and robustness.

---

> ### Author Response · Authors · 2023-11-21
> **Response to Reviewer ZWQe**
>
> We appreciate the reviewer for the positive feedback and valuable suggestions.
>
> ## **Q: Depth of Analysis**
>
> Thank you for your suggestion. We've addressed paper writing issues by thoroughly polishing our paper. This includes detailed descriptions of data collection, experimental results (models, data, parameters, and evaluation metrics), and a reorganized experiment section emphasizing key conclusions. We've also introduced additional experiments in the **updated PDF**.
>
> ## **Q: Comparative analysis**
>
> Thank you for your feedback. Following your suggestions, we conducted comparative testing on two document datasets: DocLayNet (Text Detection) and RVL-CDIP (Document Image Classification).
>
> **[1]** *DocLayNet*
>
> We investigated three settings: (a) ADoPD fine-tuning with zero-shot testing on DocLayNet. (b) ImageNet-based initialization followed by fine-tuning on DocLayNet. (c) ADoPD fine-tuned model initialization followed by fine-tuning on DocLayNet. The results are presented below (also updated in **Table 4 (b)**).
>
> | Method              | Backbone           | Zero-Shot (ADoPD) | Zero-Shot (ADoPD) | Finetune (ImageNet) | Finetune (ImageNet) | Finetune (ADoPD) | Finetune (ADoPD) |
> |---------------------|--------------------|--------------------:|-----------------------:|-------------------:|-------------------:|------------------------:|------------------------:|
> |                     |                    | **mAP**             | **mAR**                | **mAP**           | **mAR**           | **mAP**               | **mAR**               |
> | Faster R-CNN        | ResNet$_{50}$       | 0.9                 | 58.5                   | 43.0              | 55.8              | 44.5                   | 60.4                   |
> |                     | ResNet$_{101}$      | 1.0                 | 56.6                   | 46.0              | 58.5              | 47.0                   | 60.7                   |
> | Deformable-DETR     | ResNet$_{50}$       | 2.2                 | 80.4                   | 74.7              | 87.2              | 75.4                   | 88.9                   |
> |                     | ResNet$_{101}$      | 2.6                 | 79.0                   | 75.4              | 85.9              | 77.2                   | 88.1                   |
>
>
> **[2]** *RVL-CIDP*
>
> We compared CLIP vision backbones: (a) Laion pretrained backbone + RVL-CDIP fine-tuned classifier. (b) ADoPD +*Doc2Seq* pretrained CLIP backbone fine-tuned on RVL-CDIP (updated in **Figure 6 (b)** and **Table 6**).
>
> | Model                  | Type        | Letter | Form  | Email | Hw    | Ad    | SR    | SP    | SP    | FF    | NA    | Bgt   | Inv   | Prsnt | Qnr   | Rsm   | Memo  | Avg   |
> |------------------------|-------------|--------|-------|-------|-------|-------|-------|-------|-------|-------|-------|-------|-------|-------|-------|-------|-------|-------|
> | ViT$_{\text{G-14}}$     | Supervised  | 86.20  | 76.70 | 94.28 | 93.48 | 91.81 | 71.94 | 91.10 | 89.72 | 94.97 | 83.68 | 81.24 | 87.44 | 78.26 | 84.97 | 92.94 | 85.87 | 86.57 |
> | ViT$_{\text{G-14}}$+ADoPD | Supervised | 90.87  | 84.48 | 96.98 | 95.34 | 93.76 | 82.39 | 93.51 | 93.00 | 95.61 | 89.81 | 89.62 | 92.85 | 84.29 | 90.23 | 96.45 | 92.62 | 91.38 |
>
> Significant ADoPD-based improvements were observed across all types, which highlight ADoPD's advantages for finetuning or pretraining.
>
> ## **Q: Outlier Detection and Human-in-the-loop**
>
> Thank you for your suggestion. Below, we answer your question from two perspectives: outlier detection and human-in-the-loop:
>
> **[1]** *Outlier Detection*
>
> The document presents a challenge for feature extraction, combining both images and text. A potential drawback may be in feature extraction, particularly in the initial stages relying solely on CLIP trained on natural images. While introducing additional data enhances robustness, this remains an area for improvement. Additional outlier detection methods, such as textual features, will be explored in the Appendix.
>
> **[2]** *Human-in-the-loop*
>
> During the data selection, we found that sensitive images, such as those depicting violence, might present within the in-distribution. However, recognizing their undesirability, we deemed them as failure cases. Depending solely on outlier detection to identify such outliers might yield unpredictable results. Therefore, We opted for a human-in-the-loop approach for this process.
>
> We engaged diverse annotators from various countries and genders to check selected data. Providing only basic keywords without explicit sensitive references, annotators were tasked with removing documents they deemed sensitive. Ultimately, we achieved a drop percentage of 1.77%, and manual checks confirmed the presence of sensitive information in these images.

---

### Official Review · Reviewer_BGf2 · 2023-10-31

**Soundness:** 3 good
**Presentation:** 3 good
**Contribution:** 3 good
**Rating:** 6
**Confidence:** 5

**Summary:**

This paper introduces a new annotated database to enhance automatic document comprehension and, more specifically, document page decomposition. The dataset was annotated multimodally to support 4 tasks: document entity segmentation, text detection, tagging, and captioning. The annotation was done in such a way as to combine the use of large pre-trained models and human expertise. Finally, a great effort has been made to maximize the diversity of document images.

**Strengths:**

* This paper presents a very significant and interesting work on data collection, showing a great diversity. Moreover, the use of numerous pre-trained models such as ViT, CLIP, or BLIP and the fusion of all the information predicted by these models within a single prompt is very interesting and rarely presented (and even used?) in research papers.
* Even if this is not really shown in the results tables, it is clear that this dataset will certainly help research on documents, particularly thanks to the diversity of the images and the multiple annotations.

**Weaknesses:**

* While the use of 5 pre-trained models to pre-label images is quite innovative, it would also be interesting to indicate the cost of applying these different models to an image (and even to millions of images).
* I found section 4.1 a little unclear. Although efforts have been made to explain the image selection process as fully as possible, I'm not sure that this methodology can be easily reproduced from these explanations.
* In the results tables, the authors show how the different models compare once they've been trained on ADOPD. It would have been interesting to compare these models with and without fine-tuning on ADOPD to see the real impact of this dataset.
* It would have been interesting to see results on other document datasets, starting from models pre-trained on ADOPD vs. on ImageNet for example. This would have shown the real gain of pre-training on a wide variety of document images compared to a large quantity of natural scene images.
* The metrics used to evaluate the caption performance should probably be a bit more explained in the text.

**Questions:**

* This dataset would be very interesting to explore and obviously to use for pre-training. I think it would also be great to see what generalization capabilities a model trained on this dataset would have, once applied to another use case. This is not described in the paper, but do you plan to make this dataset public? If so, how do you plan to distribute it?

---

> ### Author Response · Authors · 2023-11-21
> **Response to Reviewer BGf2**
>
> We thank the reviewer for the positive feedback and suggestions.
>
> ## **Q:  Cost of applying these different models**
>
> Good question! Our method involves multiple pretrained models. The costs (model and annotation)  can be primarily categorized as follows:
>
> **[1]** *Data Filtering*
>
> Costs in this part include GPU and GPT-4 API calling expenses. In data selection, our challenge was identifying valid document images from the 175M Laion web image pool. We employed models like Document Finding Classifier, Watermark Detection, Language Detection, OCR tools, and CLIP zero-shot tagging. Except for CLIP, other models are lightweight. We utilize GPT-4 ($0.03/1k prompt tokens) for all data during collection and *Doc2Seq* & *Doc2Tag*, with costs limited to selected images.
>
> **[2]** *Data Annotation Cost*
>
> In our model-assisted data annotation approach, model inference and training occur for each batch of data sent to annotators, incurring relatively high GPU costs. However, this cost is still smaller than human labeling expenses. Additionally, periodic model refinement was observed to accelerate annotation efficiency and reduce costs. Refer to **Fig. 12** (*Appendix, Page 17*) for the annotated cost, displaying a decrease with the increase in data.
>
> ## **Q: Unclear Section 4.1 and Reproduce**
>
> The section has been revised in the updated PDF. In addition, **Sec. 3.3** provides detailed information on data collection and annotation processes, while **Sec. 4.1** covers baseline model implementation, hyperparameters, and evaluation metrics. In **Sec. 4.2**, we analyze experimental results, including new experiments in **Tables 4 & 6**. We'll provide models and annotation guidelines for replication. The revised *Appendix* includes detailed data analyses and taxonomy analysis.
>
> ## **Q: With or Without Fine-tuning on ADoPD**
>
> Great question.  in **Table 4 (a)**, we compared models fine-tuned on EntitySeg and fine-tuned on ADoPD Doc2Mask. The zero-shot performance of EntitySeg-tuned models is lower than ADoPD-tuned ones, given EntitySeg lacks document data. Despite excelling in natural image segmentation according to **Fig. 5 (a)**, EntitySeg-trained models are unfit for documents due to significant composition differences, where layout elements outweigh content significance.
>
> ## **Q: Other Document Datasets and Pre-training**
>
> In line with your suggestion, we conducted two experiments:
>
> **[1]** *DocLayNet*
>
> In **Table 4**, ADoPD-trained models achieve good recall on the DocLayNet test set, despite a low mAP. When comparing models fine-tuned on DocLayNet with pre-trained weights from ImageNet and ADoPD, the ADoPD-based models exhibit superior performance, emphasizing their generalization ability. Below, we present some results; for more details, please refer to our revised PDF.
>
> | Method              | Backbone           | Zero-Shot (ADoPD) | Zero-Shot (ADoPD) | Finetune (ImageNet) | Finetune (ImageNet) | Finetune (ADoPD) | Finetune (ADoPD) |
> |---------------------|--------------------|--------------------:|-----------------------:|-------------------:|-------------------:|------------------------:|------------------------:|
> |                     |                    | **mAP**             | **mAR**                | **mAP**           | **mAR**           | **mAP**               | **mAR**               |
> | Faster R-CNN        | ResNet$_{50}$       | 0.9                 | 58.5                   | 43.0              | 55.8              | 44.5                   | 60.4                   |
> |                     | ResNet$_{101}$      | 1.0                 | 56.6                   | 46.0              | 58.5              | 47.0                   | 60.7                   |
> | Deformable-DETR     | ResNet$_{50}$       | 2.2                 | 80.4                   | 74.7              | 87.2              | 75.4                   | 88.9                   |
> |                     | ResNet$_{101}$      | 2.6                 | 79.0                   | 75.4              | 85.9              | 77.2                   | 88.1                   |
>
> **[2]** *RVL-CDIP*
>
> We also conducted tests on RVL-CDIP, demonstrating a significant improvement when pre-training CLIP on ADoPD data and fine-tuning on RVL-CDIP, a dataset richer in data types than DocLayNet (refer to **Table 6** and **Fig. 6 (b)**).
>
> ## **Q: Other issues (Caption Evaluation and Data)**
>
> We've enhanced caption evaluation in our revised paper. In **Sec. 4.2**, we used 5K test data to assess Doc2Seq's effectiveness. GPT-4 captions achieve commendable CIDEr scores, but a notable disparity persists compared to human annotations. Models fine-tuned on *Doc2Seq* match GPT-4 on B@n but exhibit significantly lower CIDEr scores. Additionally, in **Fig. 6 (left)**, human-annotated captions are notably longer than machine-generated ones, influencing the evaluation results.
>
> We release ADoPD to support foundational models in document understanding, aiming to inspire progress in document analysis.

---

### Official Review · Reviewer_aArm · 2023-11-01

**Soundness:** 3 good
**Presentation:** 4 excellent
**Contribution:** 3 good
**Rating:** 6
**Confidence:** 4

**Summary:**

This paper introduces a new document page decomposition dataset called ADoPD, mainly focusing on document tagging, segmentation, text detection and caption. Combining automatic procedure like machine/deep learning models and human annotator, this paper creates a much large and high-quality document dataset and still maintains low cost. Additionally, this paper conduct analysis of proposed dataset.

**Strengths:**

1.	This paper collects a large amount of data (120k), which has high resolution, wide source, and diverse appearance, the high quality data is beneficial for area. It offers segment, caption annotations which are rare in other document datasets.
2.	The “hybrid data annotation” is novel and it helps to ensure the dataset is balanced and diverse. This proposed method alleviates the demand for human annotators and still maintains the high tagging quality for this dataset.
3.	This paper offers quantitative evaluation for the diversity of data distribution, further confirmed the effectiveness of the “hybrid data annotation”.

**Weaknesses:**

1.	The detection and segmentation annotation quality are not well studied, they all come from pretrained model and corrected by human annotators. It’s better to give more detail about the procedure to correct the annotation. Especially there are some other document datasets like DocBank -- the box annotation is extracted from PDF by automatic tools and it should be much more precise than detection model.
2.	Although the proposed dataset is claimed to be suitable for caption, but the caption annotation is generated by BLIP-2 and there is no human correction. BLIP-2 is mainly trained for universal visual-language task and it might not suitable for document caption.

**Questions:**

1.	The caption annotation is just generated by BLIP-2 and there are not further correction or filter. I think this model-annotation might be meaningless for the area.
2.	The section 4.2.3 Doc2Seq claims that 5000 samples are manually annotated, but after that, the paper says “utilized 80,000 samples for training, 20,000 for testing, and another 20,000 for validation“. It’s confusing how the manually annotation are collected and used, and why it has 20,000 validation and test samples – not consistent with 5,000 manual annotation.
3.	Will the dataset be open-source?

---

> ### Author Response · Authors · 2023-11-21
> **Response to Reviewer aArm**
>
> We thank the reviewer for the encouraging words and helpful comments. We address concerns, questions, and comments below in detail.
>
> ## **Q: Mask Annotation Quality and Annotation Procedure**
>
> Great question. We've updated an explanation of our data annotation process in **Sec. 3.3**. Extracting box annotations from PDF is convenient. However, PDF extraction tools (e.g., *PyMuPDF*) may not fully parse structural information for certain PDFs, such as scanned documents in PDF format. ADoPD aims to avoid bias toward purely digital PDFs, as many documents are in image format.
>
> For text detection, another reason we chose to have annotators label directly (instead of model-assisted) is that we observed that even when the model is trained with *DocLayNet* or *DocBank*, their prediction results introduce noisy errors and are not robust to real-world document images (especially visually-rich documents). This increases the cost, as annotators need to clean lots of redundant error bounding boxes. Therefore, we did not adopt *DocBank*'s method of parsing digital PDFs but instead used human annotation. Meanwhile, we outline the cost per instance of our manual annotation. The cost of box annotation is low compared to mask annotation.
>
> |  | Add Text Bbox | Modify Mask  |   Add Mask  |
> |----------------------------------|-------------- |-----------------------|--------------------------------|
> | Cost | 0.2*$x$ | 0.7*$x$  | $x$ |
>
> We can see that masks are the most time-consuming and costly part of the entire process. We present quantitative and visual results in **Fig. 12, 13, 14** (*Appendix, Page 17, 18, 19*), demonstrating that as data annotation increases, the cost decreases when fine-tuning the mask prediction models for assistance. Additionally, we will release our data annotation guidelines.
>
>
> ## **Q: Captions and Quality**
>
> We address this question from two perspectives:
>
> **[1]** *Our caption is not just generated by BLIP-2*
>
> We apologize for any misunderstandings that may have arisen due to the writing issues in the previous article. We would like to highlight that our captions are not solely based on BLIP2. Instead, they result from *Prompt-Guided Context-Aware Captioning* (updated **Sec. 3.2**). In practice, our captions are rewritten by *GPT-4* (**Eq. 1&2**) based on BLIP-2 captions, zero-shot labels (CLIP + Document taxonomy), OCR results, visual tags (using Recognize Anything Model), as well as rules. Additionally, in **Sec. A.1.1** (*Appendix, Page 14*), we provide a detailed explanation of prompt design and conduct human evaluations to validate the effectiveness of our prompt designs.
>
> **[2]** *Prompt-Guided Context-Aware Captioning Benefits Vision-Language Modeling*
>
> We attempted to collect human captions but faced challenges in controlling their quality due to variations in annotators' understanding and summarization of documents stemming from diverse educational backgrounds. Hence, we only collected a human-annotated test set (5k) for evaluation, presented in **Fig. 6 (left)** and **Table 5**. We also find that GPT-4 rewritten captions can cover multimodal information of documents and summarize effectively with the combination of human-adjusted prompts and heuristic rules.
>
> ## **Q: Doc2Seq Data Collection and Splits**
>
> For *Doc2seq* data, we utilized prompt-guided context-aware captioning on all ADoPD images, resulting in 80k/20k/20k splits.
>
> For *Doc2seq* model evaluation, we selectively annotated a subset of document images (5k) with human input. **Table 5** employs human-written document captions for evaluation. Generally, BLIP/BLIP2 zero-shot captions exhibit lower quality compared to models refined with Doc2Seq data.
>
> To further validate the effectiveness of ADoPD captions, we conducted the experiment in **Table 6** (*Please also refer to the response to Reviewer umJE regarding the validity of captions*.). Specifically, we fine-tuned CLIP based on ADoPD captions, then fine-tuned only CLIP's vision backbone on RVL-CDIP. We observed a significant improvement in CLIP's performance through refinement with ADoPD on RVL-CDIP (see **Fig. 6(b)**). This indicates that rewriting document captions is better than using the noisy captions from Laion.  Detailed analysis in **Sec. 4.2** highlights the importance of caption rewriting for handling noisy data.
>
> ## **Q：Will the dataset be open-source?**
>
> Yes, the motivation behind creating ADoPD is to accelerate developments in document understanding. ADoPD represents a small step towards establishing a large-scale, high-quality document image dataset. There are still numerous directions to explore, including the analysis of multilingual documents and sensitive data. We are releasing ADoPD to support the development of future foundational models for document understanding.

---

### Official Review · Reviewer_umJE · 2023-11-02

**Soundness:** 3 good
**Presentation:** 2 fair
**Contribution:** 3 good
**Rating:** 6
**Confidence:** 3

**Summary:**

To help the community develop techniques that process documents (like letters, forms, emails, news articles, resumes, and research papers), this work introduces a large dataset for several document understanding tasks. They annotate this data for document classification and for a few "page-decomposition" tasks, namely, Doc2Mask, Doc2Box, and Doc2Seq.

The underlying data is labeled and derived from Laion. However, the authors seem to have some concern about the diversity/balance of the types of data in there and/or the quality of the labels. Annotating this data is hard because of its scale. The authors introduce a rich pipeline to discover document taxonomies. The taxonomy developed in this way helps ensure that the final dataset is diverse and balanced. This pipeline is based on a human in the loop with information derived from several foundation models, namely, an LLM that orchestrates information from an image feature extractor, an OCR model, an image captioning model, an image tagging model, and a vision-language model.

The authors analyze the document taxonomy and find that it contributes "to a more comprehensive and balanced dataset". Then, they evaluate several models on their page-decomposition tasks, namely, Doc2Mask, Doc2Box, and Doc2Seq.

**Strengths:**

1. This is clearly a very large and well-managed effort that contributes a resource that the community will benefit from. The authors are thoughtful and thorough in working to ensure the quality of the data.

2. The proposed algorithm for hybrid data annotation is very rich and interesting. I'm not aware of any other successful orchestration of this number of models of various modalities, within a powerful human-in-the-loop iterative k-means clustering. There may be much of independent value in this process, beyond the benchmark itself.

**Weaknesses:**

1. The paper is particularly hard to follow! I cannot overstate this: it affects every section of the paper, from the abstract to the evaluation, so I'm not even 100% sure I understand the work. All sections contain a lot of unnecessary commentary (like comments on novelty or general motivations) that do not contribute a lot of value. At the same time, the background and organization are lacking. Please spend that space on background setup or technical details. For instance, half-way through the introduction, it was not really clear what Document Page Decomposition Dataset means (that's a lot of nouns in a sequence).

2. What is the exact contribution in the paper, compared to the data source in Laion? Were the labels of Doc2Mask, Doc2Box, and Doc2Seq just used as-is from existing data or labeled in this work? It's a significant weakness in my opinion that I'm not sure how to answer this.

3. The evaluation repeatedly claims that the authors "carry out experiments using various baseline models and parameter configurations, confirming the effectiveness of ADOPD for document domain". How does evaluating the models confirm the effectiveness of the dataset? It's very plausible in some ways, but could you be more explicit? How much of the data was confirmed to be of high quality and what kind of error analysis was conducted, etc., particularly for any labels (other than taxonomy, which is well-analyzed) generated automatically.

**Questions:**

See weakness 2.

---

> ### Author Response · Authors · 2023-11-21
> **Response to Reviewer umJE**
>
> We thank the reviewer for reviewing our paper and providing valuable suggestions.
>
> ## **Q: Background Setup and Technical Details**
>
> Thank you for pointing out the paper's writing issues. We've thoroughly polished it in the **updated PDF**, with key improvements highlighted below.
>
> **[1]** *Background Setup*
>
> Follow your suggestions, we added a document page decomposition task introduction at **Sec. 1&3**. To enhance clarity, **Fig. 2** was moved to the **Sec. 1** with refined words. In this adjustment, paragraph 4 aligns with **Fig. 2** to facilitate better understanding.
>
> **[2]** *Technical Details*
>
> We have improved the writing of technical details.
>
> - **Sec. 3.2**: We describe the models used in taxonomy discovery and improve the connection between **Alg. 1** and the text by including **Fig. 4** for better understanding. We add indices for each step corresponding to **Fig. 4**.
> - **Sec. 3.3**: This new section now includes previously missing data collection details. We have also added detailed collection methods (e.g., prompts, costs, etc.) and data details in the *Appendix*.
> - **Sec. 4.1 & 4.2**: We added a detailed description of the baseline models and evaluation. We supplemented **Table 4** (a, b) and **Table 6**, along with **Fig. 6**, and included two additional analyses for generalization and captions.
>
> ## **Q: Contributions**
>
> The major contributions of this paper can be categorized into two main points: the collected dataset and the proposed methods for dataset collection and evaluation.
>
> **[1]** *Data Contributions*
>
> The main contribution is our extensive document dataset, detailed in **Table 1**. Notably, *Doc2Mask* and *Doc2Box* are human-annotated, while *Doc2Seq* and *Doc2Tag* involve pre-trained models (e.g., CLIP, GPT-4) with human assistance. *Doc2Mask* is the costliest part (as for costs, please refer to the annotation cost response to Reviewer aArm). Unlike natural images with fixed object sets, understanding and identifying areas to annotate in different document images requires prior comprehension.
>
> We release the ADoPD to support foundational model development in document understanding, fostering advancements in document analysis.
>
> **[2]** *Method Contributions*
>
> Another contribution is our proposed data collection pipeline, detailed in **Sec. 3.2** and **Sec. 3.3**. Data-driven document taxonomy discovery is a cornerstone, determining the diversity and distribution of our data. To the best of our knowledge, our paper is pioneering in leveraging large-scale pretrained models for taxonomy discovery and data filtering in the document domain.
>
>
> ## **Q: Model Evaluation and Data Analysis**
>
> Thank you for your suggestions. We answer this question from two aspects:
>
> **[1]** *Data Effectiveness*
>
> ADoPD stands out with fine-grained visual entity masks and text bounding boxes. Our experiments on data effectiveness focus on evaluating mainstream models, backbones (**Table 2&3**), and assessing the generalization ability of models trained on ADoPD.
>
> In updated **Sec. 4.2**, we study the generalization ability with two new experiments: (a) Training models on ADoPD and testing on DocLayNet, results in **Table 4 (b)**. (b) Testing ADoPD fine-tuned CLIP image encoder on RVL-CDIP, results in **Fig. 6 (a)** and **Table 6**. Below, we show the RVL-CDIP results, highlighting the benefits of our data.
>
> | Model                  | Type        | Letter | Form  | Email | Hw    | Ad    | SR    | SP    | SP    | FF    | NA    | Bgt   | Inv   | Prsnt | Qnr   | Rsm   | Memo  | Avg   |
> |------------------------|-------------|--------|-------|-------|-------|-------|-------|-------|-------|-------|-------|-------|-------|-------|-------|-------|-------|-------|
> | ViT$_{\text{G-14}}$     | Supervised  | 86.20  | 76.70 | 94.28 | 93.48 | 91.81 | 71.94 | 91.10 | 89.72 | 94.97 | 83.68 | 81.24 | 87.44 | 78.26 | 84.97 | 92.94 | 85.87 | 86.57 |
> | ViT$_{\text{G-14}}$+ADoPD | Supervised | 90.87  | 84.48 | 96.98 | 95.34 | 93.76 | 82.39 | 93.51 | 93.00 | 95.61 | 89.81 | 89.62 | 92.85 | 84.29 | 90.23 | 96.45 | 92.62 | 91.38 |
>
> From above results and **Table 4,6**, we can see that benefiting from the diverse and fine-grained dense annotations in ADoPD, the trained models exhibit good generalization across various document datasets (DocLayNet and RVL-CDIP).
>
> **[2]** *Data Quality*
>
> During data annotation, ensuring data diversity and quality is the original intention of ADoPD. For human-annotated data, we ensure that annotators and reviewers are separate to allow a fair confirmation of the quality and reliability of *Doc2Mask* and *Doc2Box* annotations.
>
> Additionally, annotators from diverse backgrounds and countries were employed to assess the sensitivity of document images. With basic rules provided, they removed documents deemed sensitive. Data distribution details are in **Fig. 9** (*Appendix, Page 16*).
>
> We will provide detailed data quality statistical results (with human review) along with the annotation guidelines.

---

> > ### Comment · Reviewer_umJE · 2023-11-23
> >
> > Thank you for the detailed response. I have increased my score to 6/10.

---

> > > ### Author Response · Authors · 2023-11-23
> > > **Thank you!**
> > >
> > > Thank you for taking time to reviewing our work and discussions. We are glad that our response has addressed your concerns and appreciate your recognition of our paper.

---

### Author Response · Authors · 2023-11-21
**General Response to All**

We appreciate the time and effort invested by all the reviewers in evaluating our manuscript and providing constructive suggestions.

We are grateful for the reviewers’ remarks regarding the strengths of our work:

- Our Data: "*very large and well-managed*" (umJE); "*the high quality data is beneficial for area*" (aArm); "*very significant and interesting work on data collection*" (BGf2); "*most extensive and comprehensive database for document image segmentation and object detection*" (ZWQe).

- Our Method: "*the proposed algorithm for hybrid data annotation is very rich and interesting*" (umJE); "*the hybrid data annotation is novel and it helps to ensure the dataset is balanced and diverse.*" (aArm); "*it is clear that this dataset will certainly help research on documents, particularly thanks to the diversity of the images and the multiple annotations.*" (BGf2);  "*presents the first systematic exploration of document taxonomy using a blend of machine learning models and a human-in-the-loop approach.*" (ZWQe).

We also sincerely appreciate all reviewers' help in highlighting areas that require improvement, as their feedback has motivated us to *extensively revise and update our paper.* In our direct responses, we have provided detailed clarifications to address the questions raised by the reviewers. We hope that our responses successfully clarify their concerns, and we are available to provide further explanation if needed.

Once again, we express our heartfelt gratitude for the valuable comments and suggestions that have contributed to the enhancement of our work.

---

### Comment · Area_Chair_NydJ · 2023-11-23
**[ICLR 2024 Reviewers’ feedback] Please read authors’ responses and give your feedback**

Dear Reviewers,

Thanks again for your strong support and contribution as an ICLR 2024 reviewer.

Please check the response and other reviewers’ comments. You are encouraged to give authors your feedback after reading their responses. Thanks again for your help!

Best,

AC

---

### Meta-Review · Program_Chairs · 2023-12-12

**Metareview:**

All reviewers gave positive scores. The authors introduce a large dataset for several document understanding tasks. This dataset could benefit the community very well. The proposed algorithm for hybrid data annotation is very rich and interesting. The authors should still further revise the paper and make it easier to follow. Some parts (e.g., contributions, experiment details) should be further clarified.

**Justification For Why Not Higher Score:**

The writing can be further improved for better understanding.

**Justification For Why Not Lower Score:**

All reviewers gave positive scores and valued the authors' efforts about the dataset and algorithm.

---

### Decision · Program_Chairs · 2024-01-16

Accept (poster)